# Cross-Cultural Comparison of Urban Green Space through Crowdsourced Big Data: A Natural Language Processing and Image Recognition Approach

Shuhao Liu [1], Chang Su [2], Junhua Zhang [1], Shiro Takeda [1], Jiarui Liu [3] and Ruochen Yang [1],*

1   Graduate School of Horticulture, Chiba University, Chiba 271-8510, Japan
2   School of Architecture and Urban Planning, Huazhong University of Science and Technology, Wuhan 430074, China
3   School of Landscape Architecture, Beijing Forestry University, Beijing 100083, China
*   Correspondence: 21hd0501@student.gs.chiba-u.jp; Tel.: +81-080-9464-9110

**Abstract:** Understanding the relationship between environmental features and perceptions of urban green spaces (UGS) is crucial for UGS design and management. However, quantifying park perceptions on a large spatial and temporal scale is challenging, and it remains unclear which environmental features lead to different perceptions in cross-cultural comparisons. This study addressed this issue by collecting 11,782 valid social media comments and photos covering 36 UGSs from 2020 to 2022 using a Python 3.6-based crawler. Natural language processing and image recognition methods from Google were then utilized to quantify UGS perceptions. This study obtained 32 high-frequency feature words through sentiment analysis and quantified 17 environmental feature factors that emerged using object and scene recognition techniques for photos. The results show that users generally perceive Japanese UGSs as more positive than Chinese UGSs. Chinese UGS users prioritize plant green design and UGS user density, whereas Japanese UGS focuses on integrating specific cultural elements. Therefore, when designing and managing urban greenspace systems, local environmental and cultural characteristics must be considered to meet the needs of residents and visitors. This study offers a replicable and systematic approach for researchers investigating the utilization of UGS on a global scale.

**Keywords:** environmental features; urban green space; social media; natural language processing; image recognition; cross-cultural comparisons; cultural elements



## 1. Introduction

Urban Green Spaces (UGS) refer to green areas within cities that are accessible to the public. UGSs are natural and semi-natural spaces in cities that aim to improve the urban ecological environment and enhance residents' quality of life. According to this definition, UGSs can also include Urban Blue Spaces, encompassing parks, gardens, lakes, forests, lawns, rivers, and other water features. UGSs offer environmental, aesthetic, and recreational benefits that are closely linked to human well-being and quality of life, making them vital for people who live in, work in, and visit cities. These benefits are obtained through the utilization of UGSs [1–4]. However, the connotations of UGSs planned for different geographical and cultural contexts in cities often differ in subtle ways [5]. Local users' evaluation and perception of UGSs not only impacts their current usage of such spaces, but also shapes the direction of future UGS planning and construction in their cities [6].

The public's perception of UGS is shaped by the emotional connection between people and nature, influenced not only by the physical attributes of environmental features, such as greenery and water bodies, but also by the cultural background and preferences of users [7]. To effectively manage, maintain, and develop UGSs, it is crucial to clarify their comparative

advantages and understand the preferences of local users. However, accurately quantifying UGS perception is challenging because of its inherent intangible, subjective, and ambiguous properties [8,9]. Currently, UGS perception is still less considered in current planning and management. In particular, for some UGSs with an area of several hectares or more, their internal environment often has diverse characteristics, and it is difficult for a single-dimensional evaluation to objectively and comprehensively reveal public preferences [10]. Therefore, further research is needed to explore how users perceive different environmental features inside UGSs and the extent to which they influence public perceptions of UGSs.

Existing studies on landscape perception and preference focus on a single case, which does not consider the sociocultural influence on UGS perception [11]. However, the physical attributes of UGSs are universally common regardless of location, as viewed from an environmental perspective [12]. Furthermore, from evolutionary psychology's perspective, people possess some innate common understanding of the landscape, regardless of their cultural background [13]. Therefore, the differences revealed by comparative studies for different cultural backgrounds highlight the unique value of a regional UGS, and its findings reveal regional differences in landscape preferences [14]. Further comparative studies based on cross-cultural contexts are needed to provide more evidence on the cross-cultural understanding of UGS perceptions and preferences [15], and to reveal the universal applicability and differences in environmental features across different regional UGSs.

Existing research methods have typically been based on a single dimension (e.g., textual information) from questionnaires or web data, the respondents of which usually provide feedback on textual content [16]. For example, Talal and Gruntman examined shifts in urban nature site visitation during the COVID-19 pandemic by interviewing local university faculty and students [17]. Further, Wang and Yu employed a questionnaire to explore the relationship between air pollution control measures and visitor satisfaction in Chinese temples [18]. Qiu et al. crawled user information in exercise software to analyze the causes of spatial and temporal differences in the intensity of greenway use in Beijing [19]. Tan et al. discussed the impact of visitor experience on park accessibility under different travel modes by crawling user review text information on the Ctrip website [20]. In such prior research, questionnaires often face the problem of insufficient data volume, and comments from the web are often only text, lacking an analysis of user-posted images or other information and leading to a potential for biased results. In studies targeting large spatial scales, evaluating a site based on textual information alone appears to be dimensionally homogeneous. Replicable quantitative methods are required to measure the benefits of UGSs [21], and a large amount of field data is necessary to understand their reality and potential problems [22]. However, due to the complexity and cost of data collection, there may be insufficient data available [23]. Moreover, relying on human observation to understand human-nature interactions in studies with long periods, large sample size requirements, and large areas can be challenging [24], potentially leading to inaccurate results and limiting the in-depth understanding of urban green space systems [25]. Addressing these challenges requires the use of interdisciplinary approaches to obtain sufficient data and assess it quantitatively [26]. However, the development of computer crawler programs enables the semi-automation of collecting and processing various data, including text and images [27].

As autonomous user-generated data, online reviews from various online platforms (e.g., TripAdvisor, Ctrip, Google Maps, Weibo, Yelp, and Expedia) can be a good reflection of users' subjective thoughts, which are useful in quantitative landscape assessments, such as landscape perception and preference, cultural and ecosystem services, and environmental feature perception potential [28,29]. These crowdsourced data are often based on the first person and contain multidimensional descriptions of locations and mood states. In many cases, these online reviews contain not only text but also photographs taken by users. These voices provide multiple dimensions for the objective and democratic evaluation of local UGSs [30].

Thus, a comparative study from textual and photographic perspectives of comments can expand the current understanding of UGS perceptions and preferences. However,

owing to their unstructured and open-ended properties, quantified processing texts usually lend themselves to natural language processing (NLP) programs to transform large amounts of unstructured corpus data into structured and systematic data sets [31]. In particular, sentiment analysis functions can be used to analyze sentiment tendencies in texts and determine whether positive, negative, or neutral attitudes toward a topic or content are present [32]. For image data analysis in reviews, Computer Vision is typically used to identify objects or scenes in images to classify image information [33]. By combining NLP with Computer Image Recognition, a large amount of unstructured data can be quantitatively evaluated to achieve a refined evaluation of the UGS. Research on NLP and Computer Vision in landscape and planning research is still in its infancy [34], and research on using the two together to evaluate multi-region UGSs is even rarer.

To address this research gap, this study will use online review crowdsourcing big data, along with NLP and Computer Vision methods, to quantitatively evaluate and comparatively analyze the public's perceived attitudes toward UGSs in multiple locations. Additionally, this study will investigate the specific environmental characteristics that most influence public perceptions.

Specifically, this study aims to answer the following questions:

1. What are the dynamics of the distribution of perceptions of UGSs in online reviews?;

2. What environmental characteristics highlight positive or negative perceptions of UGSs for users?;

3. What are the similarities and differences in the preferences of environmental characteristics of UGSs among users from different cultural backgrounds?

First, the study area is selected and data are collected and analyzed, after which the results obtained through scientific calculations are presented. The use of UGSs in a cross-cultural context is then discussed, incorporating findings from previous research.

The purpose of this study is to describe the characteristics and similarities of local UGS use by people in different cultural contexts and analyze the reasons for their formation. This study aims to propose a replicable, efficient, and convenient system for assessing UGSs from multiple perspectives that is widely applicable in multicultural contexts. The proposed strategy has the potential to significantly enhance the efficiency and reliability of UGS utilization assessment, thereby benefiting urban planners and green space managers.

## 2. Study Area and Data Collection

We selected 36 UGSs from Japan and China as research cases for cross-cultural comparisons. Data crawling was based on Python version 3.8.5, and data analysis was performed using SPSS 26.0.

### 2.1. Study Area

To facilitate cross-cultural comparison, this study selected 36 UGSs located in Yokohama and Otsu in Japan, as well as Guangzhou and Changsha in China, as case studies (Table 1, Appendix A).

**Table 1.** Basic information on study areas.

| Column | Changsha | Guangzhou | Otsu | Yokohama |
|---|---|---|---|---|
| Country | China | China | Japan | Japan |
| Municipality | Hunan Province | Guangdong Province | Shiga prefecture | Kanagawa prefecture |
| Administrative area ($km^2$) | 11,820 | 7434 | 465 | 437.56 |
| Altitude range (masl), ca. | 63 | 21 | 94 | 23 |

**Table 1.** *Cont.*

| Column | Changsha | Guangzhou | Otsu | Yokohama |
|---|---|---|---|---|
| Total population (million) | 8.155 | 15.31 | 0.35 | 3.77 |
| Bio-geographical region | Indomalayan | Indomalayan | Palearctic | Palearctic |
| Landscape description | The northern, western, and southern edges of Changsha are mountainous, the southeast is dominated by hills, and the northeast is dominated by plains; mountains, hills, plains, and plains account for roughly one-fourth each. | The topography of Guangzhou is high in the northeast and low in the southwest, with mountains at the back and the sea at the front. The topography is complex, with five land types: low and medium mountains, hilly land, terraced land, alluvial plains, and mudflats. | The city area of Otsu City stretches long and narrow from north to south along the southwestern and southern shores of Lake Biwa. A mountain runs north-south on the border with Kyoto City, its western neighbor, and the city has a close relationship with that city, which faces across the mountain. | The topography of Yokohama City is divided into hills, terraces, river terraces, lowlands, and reclaimed land. The hills are mainly located in the western part of the city, running north and south of the city. |

China and Japan are two significant Asian countries that boast rich cultural traditions and have long histories. Both countries use Chinese characters as a writing system and place great value on their traditional culture and preservation [35]. The "Eight Views" is a traditional cultural concept that describes the most beautiful landscapes in each region, and it has a wide influence in both China and Japan. All cases selected for this study are included in the "Eight Views" of Chinese traditional culture. The concept of "Eight Views" originated in the Song Dynasty (960–1279) and later spread to Japan. It refers to eight specific scenic areas or landscapes that are considered particularly beautiful or noteworthy in Confucian culture [36]. The specific landscapes included in the "Eight Views" may vary depending on the era; however, they typically encompass natural landscapes, such as mountains, rivers, and forests, as well as cultural and historical landmarks. The eight scenic spots were usually nominated by prominent local literati and voted on by residents to determine the eight most popular spots. This is considered an early democratic attempt at citizen participation in the composition of UGSs, which is believed to best represent the characteristics of the local landscape [37].

The Japanese concept of "Eight Scenic Spots" has its origins in China, but it has evolved over a long period to form a scenic culture with unique Japanese characteristics. The cultural concept of Eight Views was made famous by the Eight Views of Xiaoxiang in Changsha and later spread to Japan by sea [38]. In contrast, Guangzhou served as the most important foreign trade window during the early part of Chinese history, and is often considered the last stop of traditional Chinese culture before heading out to the sea. Therefore, for the case study selection in China, Changsha (Eight Views of Xiaoxiang), which is the origin of the Eight Views culture [39], and Guangzhou (Eight Views of Yangcheng), which is the gateway of Chinese culture to the world [29], were selected. One case study in Japan is Yokohama (Kanazawa Eight Views), which has been the window of Japan's foreign cultural exchange since ancient times [40], and the other is the "Eight Views of Omi" (now located in Otsu, Shiga Prefecture), which is near Kyoto, the traditional cultural center of Japan [41].

While China and Japan have a long history of cultural exchange, their paths of modernization have diverged significantly, which has greatly influenced the development of their landscapes and the aesthetics of their people [42]. We argue that, due to the historical similarities and present-day idiosyncrasies of China and Japan, they are excellent samples for studying how sociocultural changes have shaped perceived preferences for landscape and environmental features [43,44]. The vast differences in the size of China and Japan

and the number of inhabitants in each country have resulted in significant variations in the number of UGS users in each country. This has led to differences in the landscape development process, even though both countries share the same cultural roots [45–48]. Therefore, studying UGSs in these four cities can not only reveal some consensus and local differences in landscape perception and preferences in different cultural contexts [49] but also provide insights into how changes in cultural perceptions and disparities in the number of users can shape the aesthetic mindset of residents.

*2.2. Data Collection*

The 36 UGS online reviews used in this study were collected from Weibo location check-ins and Google Maps (Figure 1). Weibo is China's leading social media platform, allowing users to share short text messages and pictures. The Weibo location check-in feature enables users to record the locations they have visited and share their thoughts and experiences through texts and pictures [50]. Google Maps provides a user review feature that enables users to post comments and ratings regarding specific locations. These comments can be used to provide other users with information about their location, such as scenery and ambiance. In addition, users can upload images to offer a more detailed description of their location [51]. We chose Weibo and Google Maps as our data sources because they can provide ample first-hand information about a specific location and are widely used to study social and cultural aspects, helping to understand people's needs and behaviors [52].

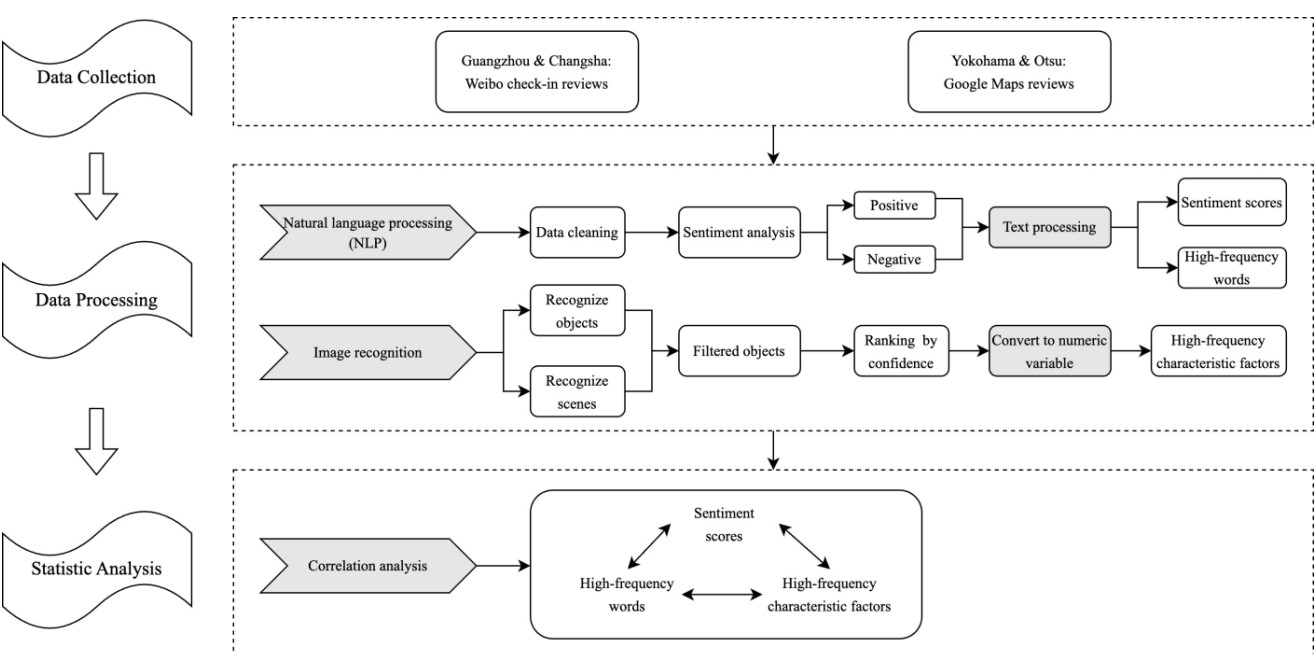

**Figure 1.** The technical framework.

This study used online review data, which was obtained with permission from Twitter and Google's licensing guidelines for non-commercial purposes. In addition, EU legislation provides mandatory exemptions for data extraction and mining for non-commercial scientific research purposes (Articles 3 and 4 of the EU Directive on Copyright in the Digital Single Market) [53]. We collected 36 UGS comments from the study participants between 1 January 2020, and 31 December 2022, where duplicate comments, comments without textual content, content without image comments, and comments identified as advertising content were automatically excluded. These comments covered multiple languages, including Chinese, English, and Japanese, and we used the Google Translate API [54] to translate and standardize the comment language to English. Finally, we collected

11,782 valid comments for analysis (2462 in Otsu, 559 in Yokohama, 4421 in Guangzhou, and 4340 in Changsha).

## 3. Methods

The collected data were quantitatively classified using natural-language sentiment analysis and image recognition. By examining high-frequency words from text comments and major scenes from user-posted photos, we analyzed the relationship between the environmental features of UGSs and user ratings using Pearson's correlation analysis to assess the perceived frequency of different environmental features (Figure 1).

### 3.1. Natural Language Processing Methods

Natural Language Processing (NLP) is a field of computer science that aims to process and understand human language, and NLP models trained with corpora can automatically perform semantic and sentiment analysis of a text. In this study, we used the Google Cloud Natural Language API [55] to perform sentiment analysis on text and extract high-frequency words to quantify the perception of UGS in-text comments. To perform sentiment analysis, we invoked the sentiment analysis function of this API on each valid comment and used a min-max normalization method to unify the sentiment scores measured by the Google NLP model to a range of 0–1. Based on the distribution of sentiment scores, we set specific thresholds to classify online comments into three sentiment polarities: positive (sentiment score $\geq$ 0.6), neutral (0.4 $\leq$ sentiment score < 0.6), and negative (sentiment score < 0.4). Simultaneously, we extracted high-frequency words from the text by invoking the lexical analysis function after data cleaning, word segmentation, word form reduction, and deactivation removal. Then, we counted the word frequency [56]. Online reviews of each UGS were quantitatively described using high-frequency words with sentiment scores.

### 3.2. Image Recognition Methods

Image recognition techniques refer to the use of computer programs to recognize objects or scenes in images, and the trained models are capable of recognizing objects in images. In this study, we called the Google Cloud Vision API [57] from the Google Vision AI series to perform object and scene recognition for each posted photo. The identified scenes in each commented photo were ranked according to the confidence level, and we retained the identified objects with a confidence level greater than 95% and converted each object into numerical variables based on their weights after recognition using the min-max normalization method and used them as the basis for further analysis and research.

### 3.3. Statistical Analysis

To explore the relationship between UGS perceptions and environmental characteristics, this study used the correlation analysis method based on SPSS 26.0. Correlation analysis is a statistical technique that measures the strength and direction of a linear relationship between two variables. In this case, it was used to assess the degree of correlation between UGS perceptions and environmental characteristics.

Pearson's correlation coefficient is a commonly used statistical measure that describes the strength of the correlation between two variables. The coefficient ranges from −1 to 1, with values closer to −1 or 1 indicating a stronger correlation and values closer to 0 indicating a weaker correlation.

In addition to the correlation coefficient, a *p*-value was calculated to determine the statistical significance of the correlation. The *p*-value measures the probability of obtaining a correlation coefficient as extreme or more extreme than that observed, assuming there is no true correlation in the population. If the *p*-value was less than 0.05, the correlation was

considered statistically significant, meaning it was unlikely to have occurred by chance. This can be expressed as follows:

$$p_{x,y} = \frac{E(XY) - E(X)E(Y)}{\sqrt{E(X^2) - (E(X))^2}\sqrt{E(Y^2) - (E(Y))^2}} \tag{1}$$

## 4. Results

### 4.1. Public Perceptions of UGS

The general sentiment towards the UGSs in China was largely negative, with 32.99% of comments possessing sentiment scores below 0.2 and 26.41% falling within the range of 0.2–0.4. On average, the sentiment score for the Chinese UGSs was 0.3869, with Guangzhou having a higher average sentiment score of 0.4215 (Figure 2) than Changsha's average score of 0.3518 (Figure 3).

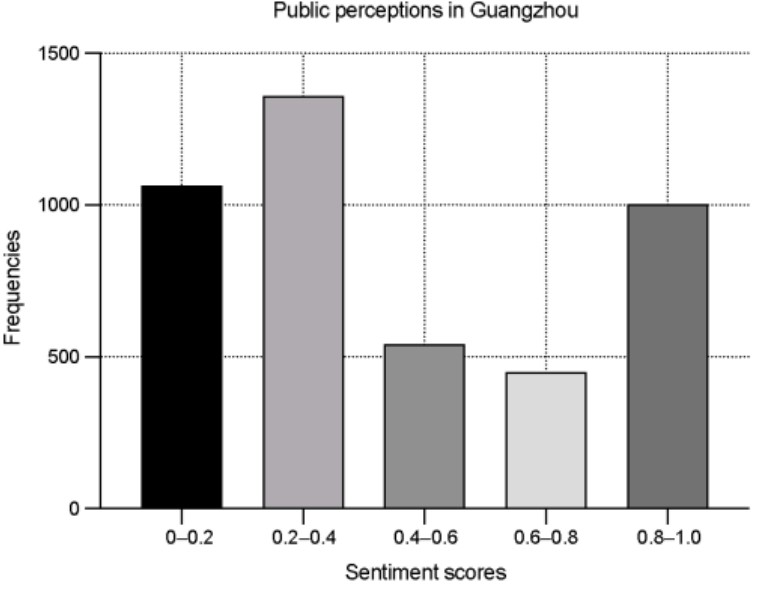

**Figure 2.** Sentiment score of UGSs in Guangzhou.

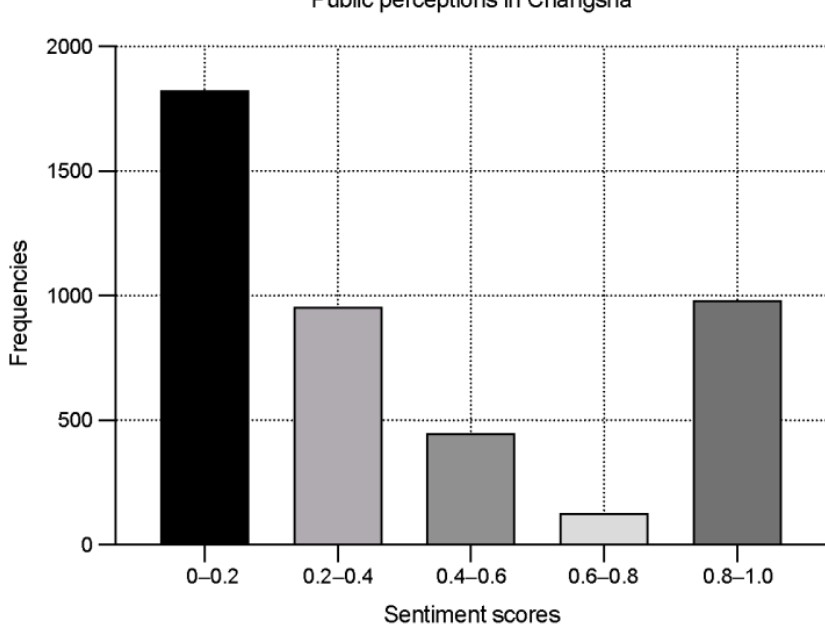

**Figure 3.** Sentiment score of UGSs in Changsha.

In contrast, the Japanese UGSs received mostly positive comments, with 57.99% of sentiment scores above 0.8 and 6.13% of scores within the range of 0.6–0.8. The average sentiment score for the UGSs in Japan was 0.6464, with Yokohama's average score being 0.6419 (Figure 4) and Otsu's average score being 0.6474 (Figure 5).

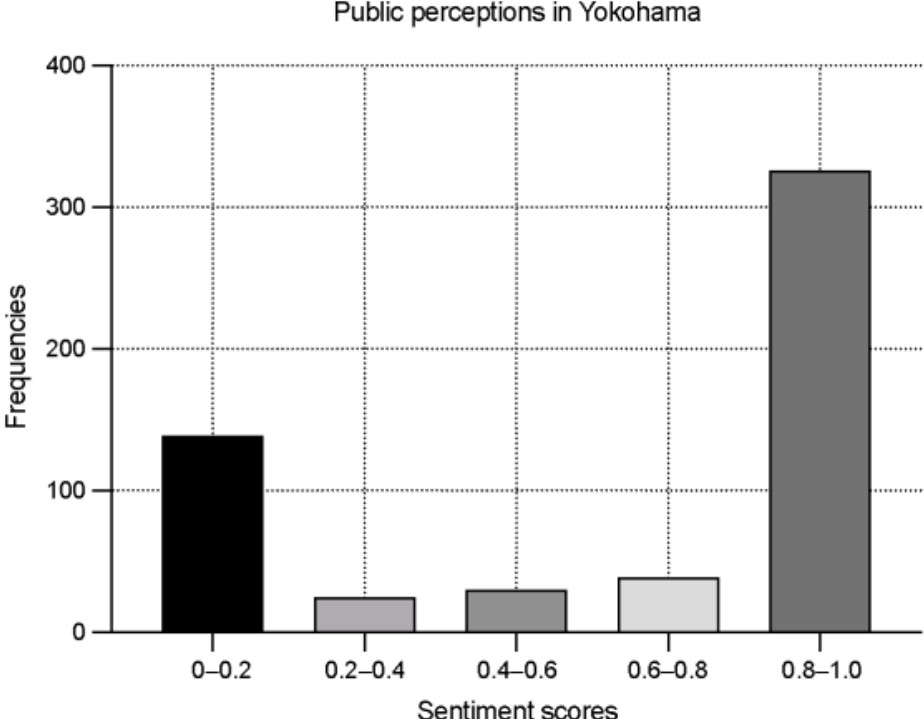

**Figure 4.** Sentiment score of UGSs in Yokohama.

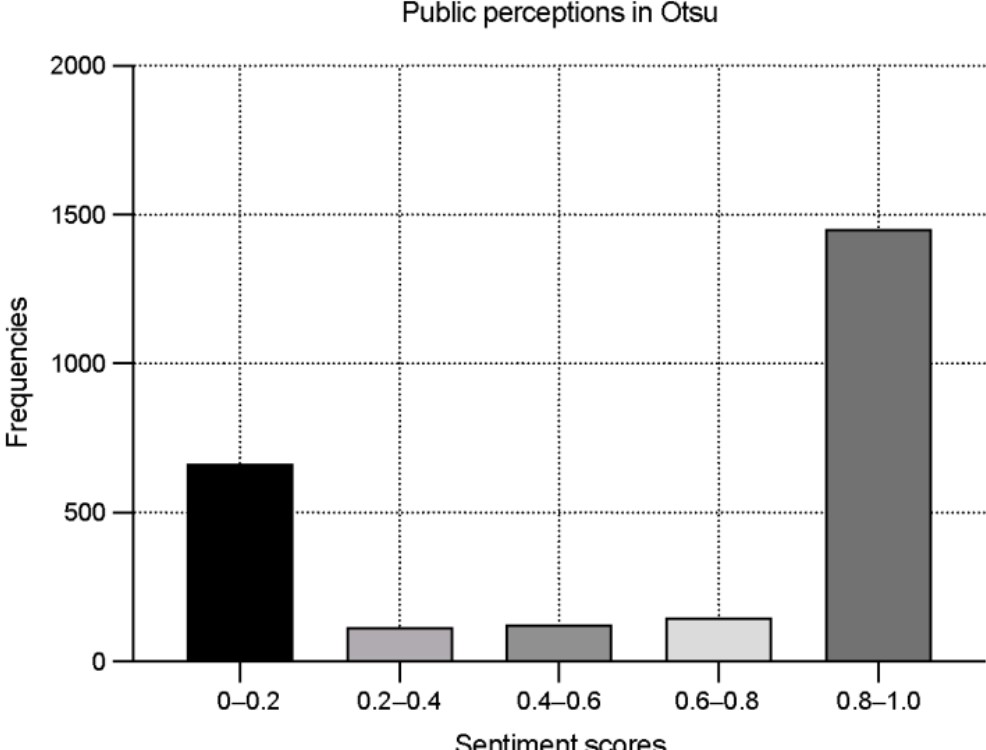

**Figure 5.** Sentiment score of UGSs in Otsu.

### 4.2. Environmental Feature Factors and High-Frequency Words in Comments

We utilized the Google Cloud Vision API to process user-posted photos and extract feature words with the highest frequencies in each UGS. The top ten feature factors were considered representative of the UGSs, while feature factors that did not make the top ten were excluded due to their low frequency (each item was less than 0.5%, and the total sum was less than 5%), which made them statistically insignificant. Our findings revealed that many similar scenes were identified in photos from around the world.

Similarly, we employed a natural language processing program to analyze the source data and obtain semantic identification objects. The results showed that the high-frequency words mentioned in the comments varied significantly among UGSs.

After analyzing the photos, we identified 17 feature factors that described the objects in each UGS user comment (Table 2). These 17 words were verified as covering more than 95% of the scenery in each UGS. They are Sky, Plant, Cloud, Water, Pond, Flower, Lip, Skin, Hairstyle, Food, Chin, Glasses, Building, Wood, Tableware, Dog, and Bird.

**Table 2.** Statistical results.

| Column | Guangzhou | | Changsha | | Yokohama | | Otsu | | TOTAL | |
|---|---|---|---|---|---|---|---|---|---|---|
| | N | % | N | % | N | % | N | % | N | % |
| Image identification objects | | | | | | | | | | |
| Sky | 9639 | 27.74 | 13,615 | 36.95 | 1076 | 31.54 | 4440 | 30.49 | 28,770 | 32.12 |
| Plant | 7468 | 21.49 | 5435 | 14.75 | 794 | 23.27 | 4895 | 33.61 | 18,592 | 20.76 |
| Cloud | 5528 | 15.91 | 8673 | 23.54 | 491 | 14.39 | 2017 | 13.85 | 16,709 | 18.65 |
| Water | 4509 | 12.98 | 4355 | 11.82 | 763 | 22.36 | 1275 | 8.75 | 10,902 | 12.17 |
| Pond | 0 | 0.00 | 0 | 0.00 | 85 | 2.49 | 103 | 0.71 | 188 | 0.21 |
| Flower | 2260 | 6.50 | 561 | 1.52 | 106 | 3.11 | 1428 | 9.80 | 4355 | 4.86 |
| Lip | 1439 | 4.14 | 595 | 1.61 | 0 | 0.00 | 0 | 0.00 | 2034 | 2.27 |
| Skin | 1302 | 3.75 | 0 | 0.00 | 0 | 0.00 | 0 | 0.00 | 1302 | 1.45 |
| Hairstyle | 956 | 2.75 | 498 | 1.35 | 0 | 0.00 | 0 | 0.00 | 1454 | 1.62 |
| Food | 906 | 2.61 | 1565 | 4.25 | 23 | 0.67 | 117 | 0.80 | 2611 | 2.91 |
| Chin | 743 | 2.14 | 0 | 0.00 | 0 | 0.00 | 0 | 0.00 | 743 | 0.83 |
| Glasses | 0 | 0.00 | 799 | 2.17 | 0 | 0.00 | 0 | 0.00 | 799 | 0.89 |
| Building | 0 | 0.00 | 750 | 2.04 | 22 | 0.64 | 100 | 0.69 | 872 | 0.97 |
| Wood | 0 | 0.00 | 0 | 0.00 | 0 | 0.00 | 124 | 0.85 | 124 | 0.14 |
| Tableware | 0 | 0.00 | 0 | 0.00 | 0 | 0.00 | 65 | 0.45 | 65 | 0.07 |
| Dog | 0 | 0.00 | 0 | 0.00 | 32 | 0.94 | 0 | 0.00 | 32 | 0.04 |
| Bird | 0 | 0.00 | 0 | 0.00 | 20 | 0.59 | 0 | 0.00 | 20 | 0.02 |
| Semantic identification objects | | | | | | | | | | |
| lake park | 806 | 21.20 | 0 | 0.00 | 0 | 0.00 | 0 | 0.00 | 806 | 7.17 |
| liwan lake park | 223 | 5.87 | 0 | 0.00 | 0 | 0.00 | 0 | 0.00 | 223 | 1.98 |
| lake Biwa | 0 | 0.00 | 0 | 0.00 | 0 | 0.00 | 228 | 9.91 | 228 | 2.03 |
| park | 385 | 10.13 | 0 | 0.00 | 34 | 6.50 | 0 | 0.00 | 419 | 3.73 |
| lake | 213 | 5.60 | 0 | 0.00 | 0 | 0.00 | 0 | 0.00 | 213 | 1.89 |
| sea | 0 | 0.00 | 0 | 0.00 | 48 | 9.18 | 0 | 0.00 | 48 | 0.43 |
| beach | 0 | 0.00 | 0 | 0.00 | 74 | 14.15 | 0 | 0.00 | 74 | 0.66 |
| pond | 0 | 0.00 | 0 | 0.00 | 39 | 7.46 | 0 | 0.00 | 39 | 0.35 |
| shrine | 0 | 0.00 | 0 | 0.00 | 32 | 6.12 | 0 | 0.00 | 32 | 0.28 |
| temple | 320 | 8.42 | 0 | 0.00 | 45 | 8.60 | 558 | 24.25 | 923 | 8.21 |
| life | 241 | 6.34 | 177 | 3.83 | 0 | 0.00 | 0 | 0.00 | 418 | 3.72 |
| heart | 0 | 0.00 | 229 | 4.96 | 0 | 0.00 | 0 | 0.00 | 229 | 2.04 |
| world | 0 | 0.00 | 197 | 4.26 | 0 | 0.00 | 0 | 0.00 | 197 | 1.75 |
| people | 146 | 3.84 | 0 | 0.00 | 66 | 12.62 | 156 | 6.78 | 368 | 3.27 |

**Table 2.** *Cont.*

| Column | Guangzhou | | Changsha | | Yokohama | | Otsu | | TOTAL | |
|---|---|---|---|---|---|---|---|---|---|---|
| | N | % | N | % | N | % | N | % | N | % |
| precincts | 0 | 0.00 | 0 | 0.00 | 0 | 0.00 | 171 | 7.43 | 171 | 1.52 |
| orange island scenic spot | 0 | 0.00 | 825 | 17.85 | 0 | 0.00 | 0 | 0.00 | 825 | 7.34 |
| national key scenic spot | 0 | 0.00 | 703 | 15.21 | 0 | 0.00 | 0 | 0.00 | 703 | 6.25 |
| orange island | 0 | 0.00 | 442 | 9.57 | 0 | 0.00 | 0 | 0.00 | 442 | 3.93 |
| place | 0 | 0.00 | 0 | 0.00 | 78 | 14.91 | 278 | 12.08 | 356 | 3.17 |
| area | 0 | 0.00 | 433 | 9.37 | 0 | 0.00 | 0 | 0.00 | 433 | 3.85 |
| orange island scenic area | 0 | 0.00 | 382 | 8.27 | 0 | 0.00 | 0 | 0.00 | 382 | 3.40 |
| scenic area | 0 | 0.00 | 295 | 6.38 | 0 | 0.00 | 0 | 0.00 | 295 | 2.62 |
| mountain scenic area | 654 | 17.20 | 0 | 0.00 | 0 | 0.00 | 0 | 0.00 | 654 | 5.81 |
| mountain | 580 | 15.26 | 0 | 0.00 | 0 | 0.00 | 0 | 0.00 | 580 | 5.16 |
| mountain peak plaza | 234 | 6.15 | 0 | 0.00 | 0 | 0.00 | 0 | 0.00 | 234 | 2.08 |
| hall | 0 | 0.00 | 0 | 0.00 | 0 | 0.00 | 183 | 7.95 | 183 | 1.63 |
| autumn leaves | 0 | 0.00 | 0 | 0.00 | 0 | 0.00 | 209 | 9.08 | 209 | 1.86 |
| cherry blossoms | 0 | 0.00 | 0 | 0.00 | 0 | 0.00 | 153 | 6.65 | 153 | 1.36 |
| tower | 0 | 0.00 | 938 | 20.30 | 0 | 0.00 | 0 | 0.00 | 938 | 8.34 |
| walk | 0 | 0.00 | 0 | 0.00 | 46 | 8.80 | 0 | 0.00 | 46 | 0.41 |
| parking lot | 0 | 0.00 | 0 | 0.00 | 61 | 11.66 | 183 | 7.95 | 244 | 2.17 |
| murasaki Shikibu | 0 | 0.00 | 0 | 0.00 | 0 | 0.00 | 182 | 7.91 | 182 | 1.62 |

Of these 17 factors, Sky, Plant, Cloud, Water, Flower, and Food were the most mentioned objects in all the UGSs, with a total proportion of 91.48% among all the screened feature factors. In the Chinese UGSs, the objects that appeared in all cities were Sky, Plant, Cloud, Water, Flower, Lip, Hairstyle, and Food, whereas, in the Japanese UGSs, the objects that appeared in all cities were Sky, Plant, Cloud, Water, Pond, Flower, Food, and Building.

It is important to note that, after comparing the original information, we found that the four feature factors, Lip, Skin, Hairstyle, and Chin, which were recognized by the images, specifically referred to the portraits taken by users. We believe that crowd participation is an important environmental feature in the utilization of UGSs in cities; therefore, our analysis considers these four factors as crowd participation in the environmental features.

After processing the textual information of the comments, we identified 32 high-frequency feature words to describe the user comments for each UGS (Table 2). These 32 words were verified to appear in more than 95% of the user comments for each UGS. These include Lake Park, Liwan Lake Park, Lake Biwa, Park, Lake, Sea, Beach, Pond, Shrine, Temple, Life, Heart, World, People, Precincts, Orange Island Scenic Spot, National Key Scenic Spot, Orange Island, Place, Area, Orange Island Scenic Area, Scenic Area, Mountain Scenic Area, Mountain, Mountain Peak Plaza, Hall, Autumn leaves, Cherry blossoms, Tower, Walk, Parking lot, and Murasaki Shikibu.

Among these high-frequency feature words, Life was the only word that appeared in all Chinese cities, accounting for 3.72% of all screened high-frequency features. The high-frequency words that appeared in all Japanese cities were Temple, People, Place, and Parking lot, accounting for 16.81% of all filtered high-frequency words.

### 4.3. Correlation Analysis

We conducted a correlation analysis between the environmental feature factors of the UGSs in each city and their sentiment scores, as well as the high-frequency feature words obtained from statistical analysis. Regarding environmental factors, we focused on the relationship between the five factors (Sky, Plant, Cloud, Water, Flower, and Food) that appeared in all UGSs and the remaining factors. For high-frequency feature words, we analyzed their correlation with environmental feature factors.

Appendix B reveals that, in the case of the UGSs in Guangzhou, the environmental factors likely to influence users' sentiment scores were Plant and Crowd participation (Lip, Skin, and Hairstyle), which show a negative correlation. The high-frequency words related

to users' sentiment scores in the comment text messages included Lake Park, Mountain Scenic Area, Temple, Life, Liwan Lake park, and Lake. Among these, Mountain Scenic Area exhibited a positive correlation, while the remainder were negatively correlated.

Regarding the environmental factors of the Guangzhou UGS, Sky, Cloud, Water, and Flower were negatively correlated with each other and with Crowd participation (Lip, Skin, Hairstyle, and Chin) and Food. Water was positively correlated with Sky, while Flower was negatively correlated with Sky, Cloud, Water, Crowd participation (Lip, Skin, Hairstyle, and Chin), and Food.

In terms of the high-frequency feature words of the Guangzhou UGS, Lake Park was negatively correlated with Sky, Cloud, Crowd participation (Lip, Skin, Hairstyle, and Chin), and Food in the environmental feature factors. Mountain Scenic Area was positively correlated with Sky, Cloud, and Crowd participation (Lip, Skin, Hairstyle, and Chin). Park was negatively correlated with Sky, Cloud, and Water in the Environmental Characteristic Factor; however, it was positively correlated with Plant and Flower. Temple was positively correlated with Sky, Plant, Flower, and Food. Life was negatively correlated with Sky and Crowd Involvement (Lip, Skin, Hairstyle, and Chin) in the Environmental Characteristic Factor but positively correlated with Flower. Mountain peak plazas were negatively correlated with Water in the Environmental Characteristic Factor, and Liwan Lake Park was negatively correlated with Sky, Cloud, and Water in the Environmental Characteristic Factor. People were negatively correlated with Plant, Water, and Flower environmental characteristics.

Appendix C shows that, for the UGSs in Changsha, several environmental characteristics can affect users' sentiment scores, including Cloud, Plant, Water, Food, Glasses, Building, Flower, and Crowd Involvement (Lip, Hairstyle). Of these, Water, Glasses, Building, and Crowd Involvement (Lip, Hairstyle) were negatively correlated with sentiment scores, while Cloud, Plant, and Food were positively correlated.

Sky was negatively correlated with Plant, Water, Food, Glasses, Building, Flower, and Crowd Involvement (Lip, Hairstyle) in the Environmental Characteristic Factor of the Changsha UGS. However, Sky was positively correlated with Cloud. Cloud was positively correlated with Plant, Food, Glasses, and Flower, but negatively correlated with Sky and Water. Plants were negatively correlated with Sky, Cloud, Water, Food, Glasses, Building, and Crowd Participation (Lip, Hairstyle) but positively correlated with Flower. Water was negatively correlated with Sky, Cloud, Plant, Food, Glasses, Flower, and Crowd Participation (Lip, Hairstyle). Flower was negatively correlated with Sky, Cloud, Water, Food, Glasses, Building, and Crowd Participation (Lip, Hairstyle) but positively correlated with the Plant.

Among the high-frequency feature words of the Changsha UGS, Tower was negatively correlated with Sky, Plant, Glasses, Building, Flower, and Crowd Participation (Lip, Hairstyle) in the Environmental Characteristic Factor. Orange Island scenic spots were negatively correlated with Cloud, Water, Flower, and Crowd Involvement (Lip, Hairstyle). National key scenic spot was negatively correlated with Plant, Water, Food, Glasses, Flower, and Crowd Involvement (Lip, Hairstyle) in the Environmental Characteristic Factor. Orange Island was negatively correlated with Sky, Plant, Glasses, and Building, while Orange Island scenic area was negatively correlated with Sky, Cloud, Water, and Flower in the Environmental Characteristic Factor. However, Scenic area was positively correlated with Sky, Cloud, Water, Food, Building, and Crowd Participation (Lip, Hairstyle) of the environmental factors. World was negatively correlated with Food, Building, and Flower in the Environmental Characteristics factor. Life was negatively correlated with Food and Glasses but positively correlated with Cloud, Plant, Water, and Flower.

In Appendix D, the environmental factors influencing user sentiment scores in Yokohama's UGSs were Sky, Plant, Water, Flower, and Building. Sky and Building were negatively correlated with sentiment scores, whereas Plant, Water, and Flower were positively correlated with sentiment scores.

In more detail, Sky was negatively correlated with Plant, Flower, Dog, and Food but positively correlated with Cloud. Plant was negatively correlated with Sky, Water, Cloud, Pond, and Food but positively correlated with Flower. Water was positively correlated with Plant, Flower, Food, and Building, and also positively correlated with Pond. Cloud was negatively correlated with Plant, Flower, and Food but positively correlated with Sky and Pond. Food was negatively correlated with Sky, Plant, Water, and Cloud.

Additionally, Beach was negatively correlated with Plant but positively correlated with Cloud, People was positively correlated with Dog, Sea was positively correlated with Food, Walk was positively correlated with Water, and Temple was positively correlated with Cloud. Temple was negatively correlated with Cloud in the Environmental Characteristic Factor but positively correlated with Plant, and Pond was positively correlated with Plant in the Environmental Characteristic Factor. Shrine was negatively correlated with Sky, Water, and Cloud in the Environmental Characteristic Factor but positively correlated with Plant, Flower, and Building.

Appendix E shows that, for Otsu's UGS, the environmental factors that affect users' sentiment scores were Plant, Wood, and Food, all of which were positively correlated with sentiment scores.

Among the environmental factors of Otsu's UGS, Plant was negatively correlated with Sky, Cloud, Water, Wood, Food, Pond, and Tableware. Sky was negatively correlated with Plant, Flower, Wood, Food, and Tableware but positively correlated with Cloud. Cloud was positively correlated with Plant, Flower, Wood, and Food, while negatively correlated with Sky and Water. Flower was negatively correlated with Sky, Cloud, Water, Food, and Pond. Water was negatively correlated with Plant, Flower, Wood, Food, and Tableware but positively correlated with Sky, Cloud, and Pond. Food was negatively correlated with Plant, Sky, Cloud, Flower, and Water but positively correlated with Tableware.

In terms of high-frequency feature words, Temple was negatively correlated with Sky, Cloud, Flower, and Water in the environmental feature factor, while Lake Biwa was negatively correlated with Plant and Flower but positively correlated with Sky, Cloud, Water, and Pond. Autumn leaves were negatively correlated with Sky, Cloud, and Water in the Environmental Characteristic Factor but positively correlated with Plant. Parking lot was positively correlated with Flower, while Hall was negatively correlated with Flower but positively correlated with Sky. Murasaki Shikibu was negatively correlated with Sky, Cloud, Flower, and Water in the Environmental Characteristic Factor but positively correlated with Plant, while Precincts was negatively correlated with Flower but positively correlated with Plant and Sky. People was negatively correlated with Sky in the Environmental Characteristic Factor but positively correlated with Wood, while Cherry blossoms was negatively correlated with Cloud and Water but positively correlated with Flower.

## 5. Discussion

Like those of related studies, the findings of this study reveal that the environmental characteristics of the UGSs in each location can influence their affective scores [58]. Furthermore, the affective scores of the Chinese UGSs were generally lower than those of the Japanese UGSs. Among the environmental characteristic factors, Sky and Cloud always appeared positively correlated, whereas Sky, Cloud, Plant, Water, and Flower always appeared negatively correlated. This suggests that Sky and Cloud can serve as indicators of the quality of the weather environment in the area [59], whereas Plant and Flower can be considered indicators of the level of planting design in the UGSs of each region [60]. Similarly, Water is an indicator of the design and maintenance of water bodies in the UGSs [61], and Plant, Flower, and Water together reflect the level of attention paid to the environment and the efforts made concerning urban landscaping in the region [62].

Moreover, the presence of commercial food and beverage services in the UGSs can be reflected by the correlation between Food and Tableware, which indicates the degree of commercial development of public green spaces in the area and the ability of the area to provide these services [63]. However, environmental characteristic factors related to Crowd

participation, such as Lip, Skin, Hairstyle, and Chin, always maintain the same correlation with other elements. Therefore, Crowd participation can be considered as a whole and used as a marker for crowd participation in the UGSs of each region [64]. This reflects the level of use of public green spaces by residents of the area and their interest in them and can be used as an indicator to assess the use and popularity of UGSs in the area.

*5.1. Environmental Characteristics of UGS in Guangzhou and Changsha*

Years of intensive land use, population growth, and ecosystem exploitation have severely degraded the urban ecosystems and ecosystem services in Guangzhou [65]. These changes have led to numerous environmental problems, including air pollution, for which the city ranked 21 out of 22 cities in Guangdong Province in both 2015 and the first half of 2016 [66]. Additionally, floods, exacerbated by climate change, pose a serious threat to human life and cause significant economic damage almost every summer. Given the detrimental effects of human activity on the environment, more sustainable and eco-friendly approaches are necessary for urban development [67]. Recognizing this need, the government has turned its attention to reconstructing urban ecosystems to improve living environments and human well-being. From 2010 to 2015, the per capita area of parks and green spaces in Guangzhou city increased by 9.95 $m^2$ [68], while in 2016, the government constructed 3000 km of green alleys and established 49 different parks, including urban, forest, and wetland parks [69]. After reviewing the evidence presented, it is reasonable to suggest that the rapid expansion of green space construction in Guangzhou may have played a role in the disregard for the quality of such projects. Our research findings support this hypothesis.

The level of planted green design and residents' use of public green spaces are the main factors affecting the sentiment scores of UGS users in Guangzhou. Local parks and temples are common topics concerning the level of plant green design. Famous parks in Guangzhou, such as Liwan Lake Park, Baiyun Mountain Scenic Area, Yuexiu Mountain, Guangxiao Temple, and South Sea Temple, are endowed with high cultural and historical values, exquisite architecture, and landscaping [70]. However, the abundance of greenery can lower sentiment scores, indicating a need for more careful design and maintenance, selection of plants suited to local conditions, and improving the quality and diversity of greenery to increase people's satisfaction and comfort with urban greenery. UGS users often focus on the number of other people using UGSs, and local mountains and rivers are frequently mentioned. Baiyun Mountain, with several hiking trails and sightseeing elevators, offers magnificent views of rolling mountains and clouds [71]. However, the number of other visitors can affect one's experience, leading to lower affective scores.

Changsha is the capital city of Hunan Province, located in central China. The city enjoys a subtropical monsoon climate with an average annual temperature of about 17 degrees Celsius and an annual rainfall of about 1400 mm [72]. Changsha is rich in natural resources, including rivers, lakes, mountains, and forests. At the same time, it is an important industrial city with a consistently high economic growth rate [73]. Changsha has a large green space coverage, including different types of green spaces such as parks, forest parks, wetland parks, and green belts [74]. As of 2019, the total area of parks in Changsha had reached 5269 hectares, with a per capita park green area of 12 square meters [75]. In addition, there are many roads, green belts, and residential green spaces in Changsha, which bring ecological, landscape, and leisure benefits to the city [76]. In the construction and management of urban green spaces, Changsha actively advocates sustainable development and promotes ecological restoration and protection work [77]. The municipal government continues to increase investment in green spaces, strengthening their protection and construction, encouraging citizens to actively participate in ecological environmental protection work, and promoting the urban greening process [77]. Green space construction and management in Changsha have made positive contributions to the city's sustainable development. It is evident that Changsha has a well-managed government-directed urban green space

(UGS) system, which leads us to believe that the local UGS experience is positive. These conclusions are supported by the results of the data analysis conducted in this study.

In Changsha, the level of plant greenery design, water landscape design and maintenance, the degree of commercial development, and residents' use of public green spaces were the main factors affecting the affective score of UGS users. When users focus on the level of plant greenery design, they typically discuss scenic areas and their influence on sentiment scores is generally positive. Orange Land, a major attraction in Changsha, features tree-lined parks, lakes, and a variety of cultural and commercial facilities to enhance visitors' experiences [78]. The Yuelu Mountain Tower, a tall tower located at the bottom of Yuelu Mountain, is another popular attraction that showcases the history, culture, and beauty of Hunan. The positive impact on sentiment scores proves that landscape shaping has a positive effect on a city's image and visitors' perception. Commercial development can also enhance the UGS user experience, as seen in Orange Island and Yuelu Mountain Tower. However, the extent to which residents use public green spaces can harm sentiment scores, indicating a need to ensure that the number of users does not exceed the optimal capacity threshold.

In conclusion, this analysis shows that the level of planting design and the density of UGS users are crucial for UGSs in Guangzhou and Changsha. The widespread presence of greenery design is not sufficient to meet users' needs, and proper design and maintenance are required. Inappropriate mounding of greenery can negatively impact users' experiences, while appropriate greenery design, quality landscape creation, and maintenance can optimize such experiences. Moreover, proper waterscape design and commercial development can enhance the UGS user experience. However, exceeding the optimal capacity threshold can decrease the overall user experience.

## 5.2. Environmental Characteristics of Yokohama and Otsu UGS

Yokohama is a coastal city located in the Kanagawa Prefecture of Japan. It has a moderate climate with an average temperature of around 16 degrees Celsius and an annual rainfall of about 1400 mm [79]. As a highly developed city, Yokohama is known for its advanced technology, international trade, and cultural diversity [80].

Despite its urbanization, Yokohama has made significant efforts to preserve and expand its green spaces. The city boasts a wide range of parks and gardens, including traditional Japanese gardens and Western-style parks [81]. One of the most popular parks in Yokohama is the Yamashita Park, a seaside park with views of the harbor and the iconic Marine Tower [82]. Another notable green space is the Sankeien Garden, a traditional Japanese-style garden with historical buildings and seasonal flora [82]. In addition to these larger parks, Yokohama has many smaller green spaces and street trees that provide shade and contribute to the city's overall greenery. The city government has implemented various measures to promote the development and maintenance of green spaces, such as the "Green Yokohama" campaign, which encourages citizens to participate in tree-planting activities and the creation of community gardens [83]. Based on the evidence gathered, it is clear that Yokohama has a comprehensive and advanced plan for the development of green spaces, which is likely to result in a positive experience for users, as confirmed by our research findings.

The factors primarily affecting the sentiment scores of Yokohama UGS users were local weather conditions, level of planting design, and the design and maintenance of the water body landscape. When UGS users focused on local weather conditions, they usually discussed the beach, and its effect on the sentiment score was usually positive, indicating that Yokohama's weather promotes users' beach use. Yokohama has a temperate maritime climate with warm, humid summers and colder winters, the temperatures of which are generally mild. The average summer temperature is approximately 26 °C, and there is often a sea breeze that brings a touch of coolness. This warm and humid climate makes Yokohama a tourist-friendly place where visitors can enjoy the beach [84], thus enhancing the satisfaction and emotional scores of UGS users. When UGS users focused on the level of plant greenery design, they usually mentioned the shrine, which has long

been an important cultural resource in the area, not only to attract domestic and foreign tourists to visit but also as a place of daily recreation for residents. These shrines are homes to many ancient and spectacular structures such as altars and spirit towers, elements that are considered an important part of traditional Japanese culture and provide visitors with a unique cultural experience. Almost every corner of the shrine is planted with a variety of flowers and greenery, creating a peaceful green paradise [85]. The results showed that planting greenery at Yokohama's shrine could effectively enhance the emotional scores of UGS users, indicating that its landscape planning was sound. When UGS users focused their attention on the design and maintenance of the water landscape, they usually referred to Walk, which indicates that viewing the water landscape is usually accompanied by a walk. This is due to the proper layout and utilization of water resources at the Yokohama UGS, which creates a diverse water landscape utilizing reasonable flow design, increasing greenery along the route, and using natural topography [86], which enhances the UGS users' enjoyment and experience, thus making it conducive to improving emotional scores.

Otsu is known for its abundant greenery and natural beauty. During the springtime, cherry blossoms can be seen in various locations throughout the city [86]. The cherry blossoms draw many visitors to the city, and Otsu holds various events and festivals to celebrate their beauty and significance [87]. The "Eight Views of Omi" refer to a set of scenic views around Lake Biwa and the surrounding mountains that have been celebrated in Japanese art and literature for centuries [88]. These views include spots such as Mount Hira, Ishiyama-dera Temple, and the Torii Gate at Shirahige Shrine.

In Otsu, visitors can experience these views firsthand by visiting various locations around the city, such as the Enryaku-ji Temple on Mount Hiei, which offers panoramic views of Lake Biwa and the surrounding mountains [d6], or the Hiyoshi Taisha Shrine, which features a torii gate that is one of the "Eight Views of Omi". The evidence collected indicates that Otsu places great importance on cultural landscapes and cherishes cherry blossoms as a significant aspect of its cultural heritage. Our research findings also support this conclusion.

The affective scores of UGS users in Otsu were mainly influenced by two factors: the level of greenery design and the degree of commercial development. In terms of the level of plant green design, cherry blossoms are a topic of great interest because they are the national flower of Japan, and Otsu is a famous cherry blossom viewing destination. Every spring, pink cherry blossoms bloom in parks, along the riverside, and along the roads in Otsu, the most famous of which is the cherry blossom path along the shore of Lake Biwa, an 8-km-long walking path lined with patches of cherry trees, which is spectacular [89]. These cherry blossom sites not only enhance the emotional score of UGS users, but they also bring important economic and tourism benefits to the city. In terms of the degree of commercial exploitation, the cherry blossom season is of significant commercial importance in Japan. Every spring, the sight of cherry blossoms in full bloom attracts many tourists to view them and take pictures, which also becomes an important marketing opportunity for businesses. Many businesses and street vendors offer a wide variety of food and souvenirs during the cherry blossom season, which not only enhances visitors' experience and desire to shop but also promotes local cultural exchange and economic development. Therefore, an increased level of local commercial development also improves the sentiment score of UGS users.

In summary, the analysis showed that both Yokohama and Otsu's UGSs have excellent levels of plant greenery. Yokohama's plant greenery is usually reflected together with other landscape elements, and local weather conditions, the level of plant greenery design, and water body landscape design and maintenance are all important factors that influence users' sentiment scores. Among them, the beach, shrine, and water body landscapes are users' preferred landscapes, and through scientific planning and design, they have become important resources of the Yokohama UGS. In contrast, Otsu's UGS landscape creation focuses on cherry blossoms and effectively enhances the user experience by designing around them.

*5.3. Limitations and Future Research*

This study utilizes a social data-based approach for analysis, but its accuracy is limited by the data collection and text processing algorithms employed. Due to the nature of this method, it may not fully capture the specific characteristics of UGS visitors, such as whether they are alone or accompanied by children or pets, which may introduce bias. Additionally, data collected from specific websites may reflect age group bias, affecting the reliability of the results.

Therefore, improvements can be made in data collection and processing in future work, and technological advancements can provide a more accurate and stable system. Future research can explore various data sources and diversified analysis methods to improve the accuracy and reliability of social data-based research.

However, this study's limitations do not make its results useless or unreliable. Social media data can still provide valuable insights into people's attitudes and behaviors towards urban green spaces and can complement other data sources, such as surveys, interviews, and observations. It is important to consider the strengths and limitations of different data sources and to use a variety of methods to gain a more comprehensive understanding of the research topic.

## 6. Conclusions

In conclusion, the UGS design and user preferences in China and Japan share similarities but also reflect distinct cultural and environmental factors. Both countries recognize the importance of incorporating greenery and natural landscapes in urban areas, although the emphasis varies. Chinese UGS design prioritizes practicality and quantity, whereas Japanese design emphasizes aesthetics and specific cultural elements. In both countries, water features are a common element in UGS design and contribute to users' positive evaluations of the environment. Nevertheless, the approaches used to create a pleasant environment for visitors differ between the two countries.

In the future, Chinese city builders should focus on designing more diverse and aesthetically pleasing landscapes and strengthening maintenance management practices to ensure the long-term health and stability of the parkland. This will enhance people's satisfaction and comfort with urban greenery, enhance the creation of cultural landscapes, and promote a sustainable environment. In contrast, in Japan, it is important to strike a balance between development and preserving the existing harmonious environment. Japanese designers can learn from China's experience and consider regulating visitor numbers when appropriate.

To summarize, both countries can learn from each other's successes and challenges in UGS design and strive to create an optimal environment for users while respecting cultural and environmental factors.

This study designs a robust, efficient, and reproducible evaluation system for UGSs. Leveraging numerous trained AI models overcomes the limitations of small data volumes and the difficulty of quantitative analysis that were common in traditional surveys. The primary advantage of this evaluation system is its ability to quickly incorporate the results of interdisciplinary AI research, thereby continuously improving the accuracy of UGS evaluations.

The evaluation system presented in this paper can be instrumental in assisting urban planners and managers in comprehending the patterns of UGS usage and making prompt and appropriate decisions. Ultimately, this system can contribute to better urban planning and management, which will lead to a healthier and more sustainable urban environment.

**Author Contributions:** Conceptualization, S.L. and R.Y.; methodology, S.L.; software, S.L.; validation, S.L., C.S. and S.T.; formal analysis, S.L.; investigation, S.T.; resources, C.S.; data curation, S.L.; writing—original draft preparation, S.L.; writing—review and editing, S.L.; visualization, J.L.; supervision, J.Z.; project administration, S.T.; funding acquisition, C.S. All authors have read and agreed to the published version of the manuscript.

**Funding:** This research was funded by JST SPRING, Grant Number JPMJSP2109 (Japan) and the Ministry of Education of Humanities and Social Science project (NO. 21YJCZH137, China).

**Data Availability Statement:** We have ensured that the Sina Weibo check-in data and Google map review data are open access to the public.

**Acknowledgments:** The authors thank Carrie Yan for technical support regarding the Python-based crawler.

**Conflicts of Interest:** The authors declare no conflict of interest.

## Appendix A. The Check-in Website of 36 UGSs

| No. | Name of UGS | Location | Check-In Page |
|---|---|---|---|
| 1 | Baiyun Mountain Scenic Area | Guangzhou | https://weibo.com/p/100101B2094654D46CAAF8409C (accessed on 5 March 2023) |
| 2 | Shanding Park | Guangzhou | https://weibo.com/p/100101B2094757D069AAF4419B (accessed on 5 March 2023) |
| 3 | Beijiang Miniature Three Gorges | Guangzhou | https://weibo.com/p/100101B2094452D265A7FC419D (accessed on 5 March 2023) |
| 4 | Guangxiao Temple | Guangzhou | https://weibo.com/p/100101B2094654D46EA3FE4898 (accessed on 5 March 2023) |
| 5 | Haizhu Lake Park | Guangzhou | https://weibo.com/p/100101B2094757D06BA5FB439F (accessed on 5 March 2023) |
| 6 | Liwan Lake Park | Guangzhou | https://weibo.com/p/100101B2094757D068A1FD4998 (accessed on 5 March 2023) |
| 7 | South Sea Temple | Guangzhou | https://weibo.com/p/100101B2094654D76AA6FE4493 (accessed on 5 March 2023) |
| 8 | Pazhou Pagoda | Guangzhou | https://weibo.com/p/100101B2094654D46EA5FC409E (accessed on 5 March 2023) |
| 9 | Five Immortals Taoist Temple | Guangzhou | https://weibo.com/p/100101B2094654D76DA0F4429D (accessed on 5 March 2023) |
| 10 | Xiqiao Mountain | Guangzhou | https://weibo.com/p/100101B2094653D36FAAFB4698 (accessed on 5 March 2023) |
| 11 | Yaozhou Ruins | Guangzhou | https://weibo.com/p/100101B2094654D664A6F8409A (accessed on 5 March 2023) |
| 12 | Yuexiu Mountain | Guangzhou | https://weibo.com/p/100101B2094757D06FA3F9409E (accessed on 5 March 2023) |
| 13 | Zhenhai Tower | Guangzhou | https://weibo.com/p/100101B2094757D06AA1FA489E (accessed on 5 March 2023) |
| 14 | Hengshan Mountain | Changsha | https://weibo.com/p/100101B209475DD069A2F5409E (accessed on 5 March 2023) |
| 15 | Huiyan Peak | Changsha | https://weibo.com/p/100101B2094757D169AAFB479F (accessed on 5 March 2023) |
| 16 | Orange Isle | Changsha | https://weibo.com/p/100101B2094757D068A3FE479F (accessed on 5 March 2023) |
| 17 | Pingzhou Academy | Changsha | https://weibo.com/p/100101B2094251D66AA3FF449A (accessed on 5 March 2023) |
| 18 | Wulingyuan Scenic Area | Changsha | https://weibo.com/p/1001018008643081100000000 (accessed on 5 March 2023) |

| No. | Name of UGS | Location | Check-In Page |
|---|---|---|---|
| 19 | Xiangyin Xiangjiang Bridge | Changsha | https://weibo.com/p/100101B2094652D46DA1F4499A (accessed on 5 March 2023) |
| 20 | Yueyang Tower | Changsha | https://weibo.com/p/100101B2094256D665ABFA419E (accessed on 5 March 2023) |
| 21 | Zhaoshan Scenic Area | Changsha | https://weibo.com/p/100101B2094757D16EA7FB439B (accessed on 5 March 2023) |
| 22 | Syoumyou Temple | Yokohama | https://www.google.com/maps/place/%E7%A7%B0%E5%90%8D%E5%AF%BA/@35.3441892,139.6282087,17z/data=!4m8!3m7!1s0x601843dd788274f5:0x4a5e0dcb461d0c43!8m2!3d35.3441892!4d139.6304027!9m1!1b1!16s%2Fg%2F121_jp5c?hl=ja (accessed on 5 March 2023) |
| 23 | Ocean Park | Yokohama | https://www.google.com/maps/place/%E6%B5%B7%E3%81%AE%E5%85%AC%E5%9C%92/@35.3385836,139.632486,17z/data=!4m8!3m7!1s0x601843d8ab7cf81b:0xa489ec4fab8d8d!8m2!3d35.3385836!4d139.63468!9m1!1b1!16s%2Fg%2F12353pcp?hl=ja (accessed on 5 March 2023) |
| 24 | Kanazawa Eight Views Park | Yokohama | https://www.google.com/maps/place/%E9%87%91%E6%B2%A2%E5%85%AB%E6%99%AF%E5%85%AC%E5%9C%92/@35.3282867,139.6229919,17z/data=!4m8!3m7!1s0x6018415c87b45c5d:0xc93ca38b6f2a5e77!8m2!3d35.3282867!4d139.6251859!9m1!1b1!16s%2Fg%2F11bz0xb_bj?hl=ja (accessed on 5 March 2023) |
| 25 | Seto Shrine | Yokohama | https://www.google.com/maps/place/%E7%80%AC%E6%88%B8%E7%A5%9E%E7%A4%BE/@35.3324974,139.6198,17z/data=!3m1!5s0x601841585bef5e75:0x920ad99c1dd40b83!4m8!3m7!1s0x601841585eb2e90b:0x2c0fd90e5d320b5f!8m2!3d35.3324974!4d139.621994!9m1!1b1!16s%2Fg%2F120n82j2?hl=ja (accessed on 5 March 2023) |
| 26 | Teko Shrine | Yokohama | https://www.google.com/maps/place/%E6%89%8B%E5%AD%90%E7%A5%9E%E7%A4%BE/@35.3392746,139.6053551,14.6z/data=!4m8!3m7!1s0x601843f6ee499a7b:0x6138536f8b4f93e7!8m2!3d35.3425218!4d139.6128589!9m1!1b1!16s%2Fg%2F1tfcc3_r?hl=ja (accessed on 5 March 2023) |
| 27 | Sunset Bridge | Yokohama | https://www.google.com/maps/place/%E5%A4%95%E7%85%A7%E6%A9%8B/@35.3259438,139.6293349,17z/data=!4m8!3m7!1s0x60184169b1492331:0x23fd6942183562e5!8m2!3d35.3259438!4d139.6315289!9m1!1b1!16s%2Fg%2F11bz0zmgqv?hl=ja (accessed on 5 March 2023) |
| 28 | Susaki Shrine | Yokohama | https://www.google.com/maps/place/%E6%B4%B2%E5%B4%8E%E7%A5%9E%E7%A4%BE/@35.3338838,139.624228,17z/data=!4m8!3m7!1s0x6018415f1bd32673:0x90addb5cec607d85!8m2!3d35.3338838!4d139.626422!9m1!1b1!16s%2Fg%2F11gzdj092?hl=ja (accessed on 5 March 2023) |

| No. | Name of UGS | Location | Check-In Page |
|---|---|---|---|
| 29 | First Nagisa Park | Otsu | https://www.google.com/maps/place/%E7%AC%AC1%E3%81%AA%E3%81%8E%E3%81%95%E5%85%AC%E5%9C%92/@35.1265264,135.9483055,17z/data=!4m8!3m7!1s0x60017436ba0e1205:0xfe139c06c63b10f6!8m2!3d35.1265264!4d135.9504995!9m1!1b1!16s%2Fg%2F1tdc9qmt?hl=ja (accessed on 5 March 2023) |
| 30 | Seta Tang Bridge | Otsu | https://www.google.com/maps/place/%E7%80%AC%E7%94%B0%E3%81%AE%E5%94%90%E6%A9%8B/@34.9729506,135.9044914,17z/data=!4m8!3m7!1s0x60016d54735df983:0x47ae03bb412bff0b!8m2!3d34.9729506!4d135.9066854!9m1!1b1!16s%2Fg%2F120q_zxw?hl=ja (accessed on 5 March 2023) |
| 31 | Mangetsu Temple | Otsu | https://www.google.com/maps/place/%E6%BA%80%E6%9C%88%E5%AF%BA%E6%B5%AE%E5%BE%A1%E5%A0%82/@35.1098513,135.9193994,17z/data=!4m8!3m7!1s0x600175404bc2e9d3:0x8d98ee97122ce359!8m2!3d35.1098513!4d135.9215934!9m1!1b1!16s%2Fm%2F0gmbk9h?hl=ja (accessed on 5 March 2023) |
| 32 | Ishiyama Temple | Otsu | https://www.google.com/maps/place/%E7%9F%B3%E5%B1%B1%E5%AF%BA/@34.9605093,135.9033826,17z/data=!4m8!3m7!1s0x60016d4ae2b177ef:0x340071f09f91d53f!8m2!3d34.9605093!4d135.9055766!9m1!1b1!16s%2Fm%2F02rxwnt?hl=ja (accessed on 5 March 2023) |
| 33 | Yabase-Kihan Island Park | Otsu | https://www.google.com/maps/place/%E7%9F%A2%E6%A9%8B%E5%B8%B0%E5%B8%86%E5%B3%B6%E5%85%AC%E5%9C%92/@35.0064381,135.9108088,17z/data=!4m8!3m7!1s0x600172dafdcf7533:0x70bd195867107bd9!8m2!3d35.0064381!4d135.9130028!9m1!1b1!16s%2Fg%2F119v7x87_?hl=ja (accessed on 5 March 2023) |
| 34 | Ōtsu Kogan Nagisa Park | Otsu | https://www.google.com/maps/place/%E5%A4%A7%E6%B4%A5%E6%B9%96%E5%B2%B8%E3%81%AA%E3%81%8E%E3%81%95%E5%85%AC%E5%9C%92/@34.9886259,135.8936671,17z/data=!4m8!3m7!1s0x60010d49719dcb7b:0x8fdc8397c6c6821c!8m2!3d34.988626!4d135.8978889!9m1!1b1!16s%2Fg%2F121_5cg8?hl=ja (accessed on 5 March 2023) |
| 35 | Karasaki Shrine | Otsu | https://www.google.com/maps/place/%E5%94%90%E5%B4%8E%E7%A5%9E%E7%A4%BE/@35.047401,135.8721001,17z/data=!4m8!3m7!1s0x60010b723f829c7d:0x705d329808b763ca!8m2!3d35.047401!4d135.8742941!9m1!1b1!16s%2Fg%2F1220924f?hl=ja (accessed on 5 March 2023) |
| 36 | Shiga-mii Temple | Otsu | https://www.google.com/maps/place/%E5%9C%92%E5%9F%8E%E5%AF%BA%EF%BC%88%E4%B8%89%E4%BA%95%E5%AF%BA%EF%BC%89/@35.0133981,135.850667,17z/data=!4m8!3m7!1s0x60010c71ef47f85b:0xd099a9b462df32dc!8m2!3d35.0133981!4d135.852861!9m1!1b1!16s%2Fg%2F12qgh7jz9?hl=ja (accessed on 5 March 2023) |

## Appendix B. Correlation between Sentiment Scores and Elements in Guangzhou

| Column | | Sentiment | Sky | Plant | Cloud | Water | Flower | Lip | Skin | Hairstyle | Food | Chin | Lake Park | Mountain Scenic Area | Mountain | Park | Temple | Life | Mountain Peak Plaza | Liwan Lake Park | Lake | People |
|---|---|---|---|---|---|---|---|---|---|---|---|---|---|---|---|---|---|---|---|---|---|---|
| Sentiment | r [1] | 1 | 0.003 | −0.044 ** | 0.024 | −0.019 | −0.019 | −0.046 ** | −0.040 ** | −0.035 * | 0.027 | 0.012 | −0.128 ** | 0.061 ** | −0.015 | −0.012 | −0.036 * | −0.038 * | −0.019 | −0.043 ** | −0.111 ** | 0.023 |
| | N [2] | 4421 | 4421 | 4421 | 4421 | 4421 | 4421 | 4421 | 4421 | 4421 | 4421 | 4421 | 4421 | 4421 | 4421 | 4421 | 4421 | 4421 | 4421 | 4421 | 4421 | 4421 |
| Sky | r | 0.003 | 1 | −0.171 ** | 0.384 ** | 0.033 * | −0.168 ** | −0.312 ** | −0.311 ** | −0.299 ** | −0.165 ** | −0.166 ** | −0.050 ** | 0.045 ** | −0.005 | −0.109 ** | 0.074 ** | −0.046 ** | 0.074 ** | −0.107 ** | 0.043 ** | −0.027 |
| | N | 4421 | 4421 | 4421 | 4421 | 4421 | 4421 | 4421 | 4421 | 4421 | 4421 | 4421 | 4421 | 4421 | 4421 | 4421 | 4421 | 4421 | 4421 | 4421 | 4421 | 4421 |
| Plant | r | −0.044 ** | −0.171 ** | 1 | −0.254 ** | −0.085 ** | 0.196 ** | −0.198 ** | −0.247 ** | −0.050 ** | −0.150 ** | −0.187 ** | 0.197 ** | −0.154 ** | −0.049 ** | 0.080 ** | 0.046 ** | −0.011 | −0.024 | −0.025 | 0.021 | −0.043 ** |
| | N | 4421 | 4421 | 4421 | 4421 | 4421 | 4421 | 4421 | 4421 | 4421 | 4421 | 4421 | 4421 | 4421 | 4421 | 4421 | 4421 | 4421 | 4421 | 4421 | 4421 | 4421 |
| Cloud | r | 0.024 | 0.384 ** | −0.254 ** | 1 | 0.029 | −0.154 ** | −0.205 ** | −0.207 ** | −0.203 ** | −0.127 ** | −0.084 ** | −0.060 ** | 0.160 ** | 0.003 | −0.121 ** | −0.003 | 0.015 | 0.053 ** | −0.079 ** | 0.176 ** | −0.016 |
| | N | 4421 | 4421 | 4421 | 4421 | 4421 | 4421 | 4421 | 4421 | 4421 | 4421 | 4421 | 4421 | 4421 | 4421 | 4421 | 4421 | 4421 | 4421 | 4421 | 4421 | 4421 |
| Water | r | −0.019 | 0.033 * | −0.085 ** | 0.029 | 1 | −0.090 ** | −0.176 ** | −0.210 ** | −0.164 ** | −0.106 ** | −0.146 ** | 0.163 ** | −0.126 ** | −0.041 ** | −0.078 ** | −0.119 ** | 0.011 | −0.104 ** | 0.108 ** | 0.091 ** | −0.062 ** |
| | N | 4421 | 4421 | 4421 | 4421 | 4421 | 4421 | 4421 | 4421 | 4421 | 4421 | 4421 | 4421 | 4421 | 4421 | 4421 | 4421 | 4421 | 4421 | 4421 | 4421 | 4421 |
| Flower | r | −0.019 | −0.168 ** | 0.196 ** | −0.154 ** | −0.090 ** | 1 | −0.029 ** | −0.116 ** | −0.099 ** | −0.068 ** | −0.076 ** | −0.028 | −0.106 ** | −0.008 | 0.060 ** | 0.097 ** | 0.219 ** | −0.018 | 0.013 | 0.017 | −0.039 ** |
| | N | 4421 | 4421 | 4421 | 4421 | 4421 | 4421 | 4421 | 4421 | 4421 | 4421 | 4421 | 4421 | 4421 | 4421 | 4421 | 4421 | 4421 | 4421 | 4421 | 4421 | 4421 |
| Lip | r | −0.046 ** | −0.312 ** | −0.198 ** | −0.205 ** | −0.176 ** | −0.029 ** | 1 | 0.564 ** | 0.244 ** | −0.015 | 0.329 ** | −0.085 ** | 0.099 ** | 0.022 | 0.137 | −0.046 ** | −0.037 * | −0.020 | 0.045 | −0.079 ** | 0.080 ** |
| | N | 4421 | 4421 | 4421 | 4421 | 4421 | 4421 | 4421 | 4421 | 4421 | 4421 | 4421 | 4421 | 4421 | 4421 | 4421 | 4421 | 4421 | 4421 | 4421 | 4421 | 4421 |
| Skin | r | −0.040 ** | −0.311 ** | −0.247 ** | −0.207 ** | −0.210 ** | −0.116 ** | 0.564 ** | 1 | 0.329 ** | −0.004 | 0.391 ** | −0.099 ** | 0.195 ** | 0.039 | −0.010 | −0.064 ** | −0.046 ** | −0.024 | 0.006 | −0.073 ** | 0.103 ** |
| | N | 4421 | 4421 | 4421 | 4421 | 4421 | 4421 | 4421 | 4421 | 4421 | 4421 | 4421 | 4421 | 4421 | 4421 | 4421 | 4421 | 4421 | 4421 | 4421 | 4421 | 4421 |
| Hairstyle | r | −0.035 * | −0.299 ** | −0.050 ** | −0.203 ** | −0.164 ** | −0.099 ** | 0.244 ** | 0.329 ** | 1 | −0.055 ** | 0.063 ** | −0.170 ** | 0.084 ** | 0.033 | 0.005 | −0.035 * | −0.054 ** | −0.031 | 0.051 | −0.067 ** | 0.033 * |
| | N | 4421 | 4421 | 4421 | 4421 | 4421 | 4421 | 4421 | 4421 | 4421 | 4421 | 4421 | 4421 | 4421 | 4421 | 4421 | 4421 | 4421 | 4421 | 4421 | 4421 | 4421 |
| Food | r | 0.027 | −0.165 ** | −0.150 ** | −0.127 ** | −0.106 ** | −0.068 ** | −0.015 | −0.004 | −0.055 ** | 1 | 0.001 | −0.077 ** | −0.031 * | 0.046 ** | −0.021 | 0.035 * | −0.010 | 0.004 | 0.025 | −0.048 ** | −0.019 |
| | N | 4421 | 4421 | 4421 | 4421 | 4421 | 4421 | 4421 | 4421 | 4421 | 4421 | 4421 | 4421 | 4421 | 4421 | 4421 | 4421 | 4421 | 4421 | 4421 | 4421 | 4421 |
| Chin | r | 0.012 | −0.166 ** | −0.187 ** | −0.084 ** | −0.146 ** | −0.076 ** | 0.329 ** | 0.391 ** | 0.063 ** | 0.001 | 1 | −0.069 ** | 0.148 ** | 0.093 | −0.03 | −0.043 ** | −0.040 ** | −0.023 | 0.007 | −0.056 ** | 0.144 ** |
| | N | 4421 | 4421 | 4421 | 4421 | 4421 | 4421 | 4421 | 4421 | 4421 | 4421 | 4421 | 4421 | 4421 | 4421 | 4421 | 4421 | 4421 | 4421 | 4421 | 4421 | 4421 |
| lake Park | r | −0.128 ** | −0.050 ** | 0.197 ** | −0.060 ** | 0.163 ** | −0.028 | −0.085 ** | −0.099 ** | −0.170 ** | −0.077 ** | −0.069 ** | 1 | −0.141 ** | −0.126 ** | −0.103 ** | −0.089 ** | −0.011 | −0.079 ** | −0.077 ** | 0.194 ** | −0.051 ** |
| | N | 4421 | 4421 | 4421 | 4421 | 4421 | 4421 | 4421 | 4421 | 4421 | 4421 | 4421 | 4421 | 4421 | 4421 | 4421 | 4421 | 4421 | 4421 | 4421 | 4421 | 4421 |
| mountain Scenic Area | r | 0.061 ** | 0.045 ** | −0.154 ** | 0.160 ** | −0.126 ** | −0.106 ** | 0.099 ** | 0.195 ** | 0.084 ** | −0.031 * | 0.148 ** | −0.141 ** | 1 | −0.097 ** | −0.092 ** | −0.080 ** | −0.043 ** | −0.071 ** | −0.069 ** | −0.068 ** | −0.049 ** |
| | N | 4421 | 4421 | 4421 | 4421 | 4421 | 4421 | 4421 | 4421 | 4421 | 4421 | 4421 | 4421 | 4421 | 4421 | 4421 | 4421 | 4421 | 4421 | 4421 | 4421 | 4421 |
| mountain | r | −0.015 | −0.005 | −0.049 ** | 0.003 | −0.041 ** | −0.008 | 0.022 | 0.039 | 0.033 | 0.046 ** | 0.093 | −0.126 ** | −0.097 ** | 1 | −0.078 ** | −0.070 ** | −0.034 * | −0.027 | −0.062 ** | −0.036 * | 0.166 ** |
| | N | 4421 | 4421 | 4421 | 4421 | 4421 | 4421 | 4421 | 4421 | 4421 | 4421 | 4421 | 4421 | 4421 | 4421 | 4421 | 4421 | 4421 | 4421 | 4421 | 4421 | 4421 |
| park | r | −0.012 | −0.109 ** | 0.080 ** | −0.121 ** | −0.078 ** | 0.060 ** | 0.137 | −0.010 | 0.005 | −0.021 | −0.03 | −0.103 ** | −0.092 ** | −0.078 ** | 1 | −0.058 ** | −0.032 * | −0.052 ** | −0.051 ** | −0.050 ** | −0.026 |
| | N | 4421 | 4421 | 4421 | 4421 | 4421 | 4421 | 4421 | 4421 | 4421 | 4421 | 4421 | 4421 | 4421 | 4421 | 4421 | 4421 | 4421 | 4421 | 4421 | 4421 | 4421 |
| temple | r | −0.036 * | 0.074 ** | 0.046 ** | −0.003 | −0.119 ** | 0.097 ** | −0.046 ** | −0.064 ** | −0.035 * | 0.035 * | −0.043 ** | −0.089 ** | −0.080 ** | −0.070 ** | −0.058 ** | 1 | −0.019 | −0.045 ** | −0.044 ** | −0.043 ** | −0.007 |
| | N | 4421 | 4421 | 4421 | 4421 | 4421 | 4421 | 4421 | 4421 | 4421 | 4421 | 4421 | 4421 | 4421 | 4421 | 4421 | 4421 | 4421 | 4421 | 4421 | 4421 | 4421 |
| life | r | −0.038 * | −0.046 ** | −0.011 | 0.015 | 0.011 | 0.219 ** | −0.037 * | −0.046 ** | −0.054 ** | −0.010 | −0.040 ** | −0.011 | −0.043 ** | −0.034 * | −0.032 * | −0.019 | 1 | 0.007 | −0.031 * | 0.140 ** | −0.027 |
| | N | 4421 | 4421 | 4421 | 4421 | 4421 | 4421 | 4421 | 4421 | 4421 | 4421 | 4421 | 4421 | 4421 | 4421 | 4421 | 4421 | 4421 | 4421 | 4421 | 4421 | 4421 |
| mountain Peak Plaza | r | −0.019 | 0.074 ** | −0.024 | 0.053 ** | −0.104 ** | −0.018 | −0.020 | −0.024 | −0.031 | 0.004 | −0.023 | −0.079 ** | −0.071 ** | −0.027 | −0.052 ** | −0.045 ** | 0.007 | 1 | −0.039 ** | −0.038 * | −0.015 |
| | N | 4421 | 4421 | 4421 | 4421 | 4421 | 4421 | 4421 | 4421 | 4421 | 4421 | 4421 | 4421 | 4421 | 4421 | 4421 | 4421 | 4421 | 4421 | 4421 | 4421 | 4421 |
| liwan Lake Park | r | −0.043 ** | −0.107 ** | −0.025 | −0.079 ** | 0.108 ** | 0.013 | 0.045 | 0.006 | 0.051 | 0.025 | 0.007 | −0.077 ** | −0.069 ** | −0.062 ** | −0.051 ** | −0.044 ** | −0.031 * | −0.039 ** | 1 | −0.037 * | −0.014 |
| | N | 4421 | 4421 | 4421 | 4421 | 4421 | 4421 | 4421 | 4421 | 4421 | 4421 | 4421 | 4421 | 4421 | 4421 | 4421 | 4421 | 4421 | 4421 | 4421 | 4421 | 4421 |
| lake | r | −0.111 ** | 0.043 ** | 0.021 | 0.176 ** | 0.091 ** | 0.017 | −0.079 ** | −0.073 ** | −0.067 ** | −0.048 ** | −0.056 ** | 0.194 ** | −0.068 ** | −0.036 * | −0.050 ** | −0.043 ** | 0.140 ** | −0.038 * | −0.037 * | 1 | −0.030 * |
| | N | 4421 | 4421 | 4421 | 4421 | 4421 | 4421 | 4421 | 4421 | 4421 | 4421 | 4421 | 4421 | 4421 | 4421 | 4421 | 4421 | 4421 | 4421 | 4421 | 4421 | 4421 |
| people | r | 0.023 | −0.027 | −0.043 ** | −0.016 | −0.062 ** | −0.039 ** | 0.080 ** | 0.103 ** | 0.033 * | −0.019 | 0.144 ** | −0.051 ** | −0.049 ** | 0.166 ** | −0.026 | −0.007 | −0.027 | −0.015 | −0.014 | −0.030 * | 1 |
| | N | 4421 | 4421 | 4421 | 4421 | 4421 | 4421 | 4421 | 4421 | 4421 | 4421 | 4421 | 4421 | 4421 | 4421 | 4421 | 4421 | 4421 | 4421 | 4421 | 4421 | 4421 |

[1] The 'r' is short for Pearson product-moment correlation coefficient. [2] The 'N' is short for number of data. The number marked by * and ** means test value is less than 0.05 and 0.01, with significant correlation.

## Appendix C. Correlation between Sentiment Scores and Elements in Changsha

| Column | | Sentiment | Sky | Cloud | Plant | Water | Food | Glasses | Building | Lip | Flower | Hairstyle | Tower | Orange Island Scenic Spot | National Key Scenic Spot | Orange Island | Area | Orange Island Scenic Area | Scenic Area | Heart | World | Life |
|---|---|---|---|---|---|---|---|---|---|---|---|---|---|---|---|---|---|---|---|---|---|---|
| Sentiment | r [1] | 1 | 0.014 | 0.081 | 0.073 ** | −0.046 ** | 0.105 ** | −0.054 ** | −0.075 ** | −0.049 ** | 0.127 ** | −0.131 ** | −0.068 ** | −0.170 ** | −0.104 ** | −0.157 ** | 0.158 ** | −0.122 ** | −0.098 ** | 0.442 ** | 0.285 ** | 0.289 ** |
| | N [2] | 4340 | 4340 | 4340 | 4340 | 4340 | 4340 | 4340 | 4340 | 4340 | 4340 | 4340 | 4340 | 4340 | 4340 | 4340 | 4340 | 4340 | 4340 | 4340 | 4340 | 4340 |
| Sky | r | 0.014 | 1 | 0.239 ** | −0.086 ** | −0.159 ** | −0.178 ** | −0.191 ** | −0.080 ** | −0.236 ** | −0.188 ** | −0.307 ** | −0.242 ** | 0.020 | 0.077 ** | 0.067 ** | 0.246 ** | −0.110 ** | −0.117 ** | 0.016 | 0.010 | −0.031 * |
| | N | 4340 | # NAME? | 4340 | 4340 | 4340 | 4340 | 4340 | 4340 | 4340 | 4340 | 4340 | 4340 | 4340 | 4340 | 4340 | 4340 | 4340 | 4340 | 4340 | 4340 | 4340 |
| Cloud | r | 0.081 | 0.239 ** | 1 | −0.356 ** | 0.181 ** | −0.327 ** | −0.144 ** | −0.003 | −0.212 ** | −0.122 ** | −0.244 ** | 0.097 ** | −0.059 ** | 0.150 ** | −0.003 | 0.113 ** | −0.122 ** | −0.101 ** | 0.041 ** | 0.025 | 0.041 ** |
| | N | 4340 | 4340 | 4340 | 4340 | 4340 | 4340 | 4340 | 4340 | 4340 | 4340 | 4340 | 4340 | 4340 | 4340 | 4340 | 4340 | 4340 | 4340 | 4340 | 4340 | 4340 |
| Plant | r | 0.073 ** | −0.086 ** | −0.356 ** | 1 | −0.088 ** | −0.094 ** | −0.083 ** | −0.101 ** | −0.074 ** | 0.385 ** | −0.147 ** | −0.149 ** | 0.080 ** | −0.122 ** | 0.232 ** | −0.121 ** | −0.007 | 0.234 ** | 0.129 ** | 0.025 | 0.057 ** |
| | N | 4340 | 4340 | 4340 | 4340 | 4340 | 4340 | 4340 | 4340 | 4340 | 4340 | 4340 | 4340 | 4340 | 4340 | 4340 | 4340 | 4340 | 4340 | 4340 | 4340 | 4340 |
| Water | r | −0.046 ** | −0.159 ** | 0.181 ** | −0.088 ** | 1 | −0.228 ** | −0.035 * | 0.062 ** | −0.124 ** | −0.067 ** | −0.171 ** | 0.379 ** | −0.055 ** | −0.172 ** | −0.054 ** | −0.131 ** | −0.097 ** | −0.030 * | −0.034 ** | 0.025 | 0.041 ** |
| | N | 4340 | 4340 | 4340 | 4340 | 4340 | 4340 | 4340 | 4340 | 4340 | 4340 | 4340 | 4340 | 4340 | 4340 | 4340 | 4340 | 4340 | 4340 | 4340 | 4340 | 4340 |
| Food | r | 0.105 ** | −0.178 ** | −0.327 ** | −0.094 ** | −0.228 ** | 1 | −0.036 * | 0.007 | −0.026 | −0.090 ** | −0.067 | 0.038 * | 0.122 ** | −0.105 ** | −0.132 ** | −0.028 | 0.009 | −0.067 ** | −0.079 ** | −0.035 * | −0.036 * |
| | N | 4340 | 4340 | 4340 | 4340 | 4340 | 4340 | 4340 | 4340 | 4340 | 4340 | 4340 | 4340 | 4340 | 4340 | 4340 | 4340 | 4340 | 4340 | 4340 | 4340 | 4340 |
| Glasses | r | −0.054 ** | −0.191 ** | −0.144 ** | −0.083 ** | −0.035 * | −0.036 * | 1 | 0.032 * | 0.431 ** | −0.074 ** | 0.067 ** | −0.128 ** | 0.105 ** | −0.037 * | 0.241 ** | −0.059 ** | 0.373 ** | −0.029 | −0.052 ** | 0.005 | −0.040 ** |
| | N | 4340 | 4340 | 4340 | 4340 | 4340 | 4340 | 4340 | 4340 | 4340 | 4340 | 4340 | 4340 | 4340 | 4340 | 4340 | 4340 | 4340 | 4340 | 4340 | 4340 | 4340 |
| Building | r | −0.075 ** | −0.080 ** | −0.003 | −0.101 ** | 0.062 ** | 0.007 | 0.032 * | 1 | −0.028 | −0.072 ** | −0.080 ** | −0.121 ** | 0.387 ** | −0.001 | 0.044 ** | −0.072 ** | 0.118 ** | −0.057 ** | −0.054 ** | −0.041 ** | −0.023 |
| | N | 4340 | 4340 | 4340 | 4340 | 4340 | 4340 | 4340 | 4340 | 4340 | 4340 | 4340 | 4340 | 4340 | 4340 | 4340 | 4340 | 4340 | 4340 | 4340 | 4340 | 4340 |
| Lip | r | −0.049 ** | −0.236 ** | −0.212 ** | −0.074 ** | −0.124 ** | −0.026 | 0.431 ** | −0.028 | 1 | −0.062 ** | 0.135 ** | −0.073 ** | −0.033 * | −0.016 * | 0.021 | −0.048 ** | 0.369 ** | −0.031 * | −0.051 ** | −0.034 | −0.013 |
| | N | 4340 | 4340 | 4340 | 4340 | 4340 | 4340 | 4340 | 4340 | 4340 | 4340 | 4340 | 4340 | 4340 | 4340 | 4340 | 4340 | 4340 | 4340 | 4340 | 4340 | 4340 |
| Flower | r | 0.127 ** | −0.188 ** | −0.122 ** | 0.385 ** | −0.067 ** | −0.090 ** | −0.074 ** | −0.072 ** | −0.062 ** | 1 | −0.062 ** | −0.089 ** | −0.102 ** | −0.082 ** | −0.080 ** | −0.022 | −0.066 ** | 0.571 ** | 0.300 ** | 0.050 ** | 0.079 ** |
| | N | 4340 | 4340 | 4340 | 4340 | 4340 | 4340 | 4340 | 4340 | 4340 | 4340 | 4340 | 4340 | 4340 | 4340 | 4340 | 4340 | 4340 | 4340 | 4340 | 4340 | 4340 |
| Hairstyle | r | −0.131 ** | −0.307 ** | −0.244 ** | −0.147 ** | −0.171 ** | −0.067 | 0.067 ** | −0.080 ** | 0.135 ** | −0.062 ** | 1 | −0.260 ** | −0.104 ** | −0.088 ** | −0.086 | −0.063 ** | 0.125 ** | −0.053 ** | −0.042 ** | −0.019 | −0.033 |
| | N | 4340 | 4340 | 4340 | 4340 | 4340 | 4340 | 4340 | 4340 | 4340 | 4340 | 4340 | 4340 | 4340 | 4340 | 4340 | 4340 | 4340 | 4340 | 4340 | 4340 | 4340 |
| tower | r | −0.068 ** | −0.242 ** | 0.097 ** | −0.149 ** | 0.379 ** | 0.038 * | −0.128 ** | −0.121 ** | −0.073 ** | −0.089 ** | −0.260 ** | 1 | −0.189 ** | −0.170 ** | −0.143 ** | −0.124 ** | −0.123 ** | −0.103 ** | −0.061 ** | −0.023 | −0.006 |
| | N | 4340 | 4340 | 4340 | 4340 | 4340 | 4340 | 4340 | 4340 | 4340 | 4340 | 4340 | 4340 | 4340 | 4340 | 4340 | 4340 | 4340 | 4340 | 4340 | 4340 | 4340 |
| orange island scenic spot | r | −0.170 ** | 0.020 | −0.059 ** | 0.080 ** | −0.055 ** | 0.122 ** | 0.105 ** | 0.387 ** | −0.033 * | −0.102 ** | −0.104 ** | −0.189 ** | 1 | −0.174 ** | 0.529 ** | −0.127 ** | −0.126 ** | −0.106 ** | −0.087 ** | −0.058 ** | −0.053 ** |
| | N | 4340 | 4340 | 4340 | 4340 | 4340 | 4340 | 4340 | 4340 | 4340 | 4340 | 4340 | 4340 | 4340 | 4340 | 4340 | 4340 | 4340 | 4340 | 4340 | 4340 | 4340 |
| national key scenic spot | r | −0.104 ** | 0.077 ** | 0.150 ** | −0.122 ** | −0.172 ** | −0.105 ** | −0.037 * | −0.001 | −0.016 * | −0.082 ** | −0.088 ** | −0.170 ** | −0.174 ** | 1 | −0.132 ** | −0.114 ** | −0.113 ** | −0.095 ** | −0.063 ** | −0.042 ** | −0.028 |
| | N | 4340 | 4340 | 4340 | 4340 | 4340 | 4340 | 4340 | 4340 | 4340 | 4340 | 4340 | 4340 | 4340 | 4340 | 4340 | 4340 | 4340 | 4340 | 4340 | 4340 | 4340 |
| orange island | r | −0.157 ** | 0.067 ** | −0.003 | 0.232 ** | −0.054 ** | −0.132 ** | 0.241 ** | 0.044 ** | 0.021 | −0.080 ** | −0.086 | −0.143 ** | 0.529 ** | −0.132 ** | 1 | −0.096 ** | −0.091 ** | −0.080 ** | −0.066 ** | −0.044 ** | −0.045 ** |
| | N | 4340 | 4340 | 4340 | 4340 | 4340 | 4340 | 4340 | 4340 | 4340 | 4340 | 4340 | 4340 | 4340 | 4340 | 4340 | 4340 | 4340 | 4340 | 4340 | 4340 | 4340 |
| area | r | 0.158 ** | 0.246 ** | 0.113 ** | −0.121 ** | −0.131 ** | −0.028 | −0.059 ** | −0.072 ** | −0.048 ** | −0.022 | −0.063 ** | −0.124 ** | −0.127 ** | −0.114 ** | −0.096 ** | 1 | −0.083 ** | −0.067 ** | 0.020 | −0.030 * | −0.033 * |
| | N | 4340 | 4340 | 4340 | 4340 | 4340 | 4340 | 4340 | 4340 | 4340 | 4340 | 4340 | 4340 | 4340 | 4340 | 4340 | 4340 | 4340 | 4340 | 4340 | 4340 | 4340 |
| orange island scenic area | r | −0.122 ** | −0.110 ** | −0.122 ** | −0.007 | −0.097 ** | 0.009 | 0.373 ** | 0.118 ** | 0.369 ** | −0.066 ** | 0.125 ** | −0.123 ** | −0.126 ** | −0.113 ** | −0.091 ** | −0.083 ** | 1 | −0.069 ** | −0.054 ** | −0.038 * | −0.037 * |
| | N | 4340 | 4340 | 4340 | 4340 | 4340 | 4340 | 4340 | 4340 | 4340 | 4340 | 4340 | 4340 | 4340 | 4340 | 4340 | 4340 | 4340 | 4340 | 4340 | 4340 | 4340 |
| scenic area | r | −0.098 ** | −0.117 ** | −0.101 ** | 0.234 ** | −0.030 * | −0.067 ** | −0.029 | −0.057 ** | −0.031 * | 0.571 ** | −0.053 ** | −0.103 ** | −0.106 ** | −0.095 ** | −0.080 ** | −0.067 ** | −0.069 ** | 1 | −0.048 ** | −0.029 | −0.019 |
| | N | 4340 | 4340 | 4340 | 4340 | 4340 | 4340 | 4340 | 4340 | 4340 | 4340 | 4340 | 4340 | 4340 | 4340 | 4340 | 4340 | 4340 | 4340 | 4340 | 4340 | 4340 |
| heart | r | 0.442 ** | 0.016 | 0.041 ** | 0.129 ** | −0.034 * | −0.079 ** | −0.052 ** | −0.054 ** | −0.051 ** | 0.300 ** | −0.042 ** | −0.061 ** | −0.087 ** | −0.063 ** | −0.066 ** | 0.020 | −0.054 ** | −0.048 ** | 1 | 0.132 ** | 0.158 ** |
| | N | 4340 | 4340 | 4340 | 4340 | 4340 | 4340 | 4340 | 4340 | 4340 | 4340 | 4340 | 4340 | 4340 | 4340 | 4340 | 4340 | 4340 | 4340 | 4340 | 4340 | 4340 |
| world | r | 0.285 ** | 0.010 | 0.025 | 0.025 | 0.025 | −0.035 * | 0.005 | −0.041 ** | −0.034 | 0.050 ** | −0.019 | −0.023 | −0.058 ** | −0.042 ** | −0.044 ** | −0.030 * | −0.038 * | −0.029 | 0.132 ** | 1 | 0.127 ** |
| | N | 4340 | 4340 | 4340 | 4340 | 4340 | 4340 | 4340 | 4340 | 4340 | 4340 | 4340 | 4340 | 4340 | 4340 | 4340 | 4340 | 4340 | 4340 | 4340 | 4340 | 4340 |
| life | r | 0.289 ** | −0.031 * | 0.041 ** | 0.057 ** | 0.041 ** | −0.036 * | −0.040 ** | −0.023 | −0.013 | 0.079 ** | −0.033 | −0.006 | −0.053 ** | −0.028 | −0.045 ** | −0.033 * | −0.037 * | −0.019 | 0.158 ** | 0.127 ** | 1 |
| | N | 4340 | 4340 | 4340 | 4340 | 4340 | 4340 | 4340 | 4340 | 4340 | 4340 | 4340 | 4340 | 4340 | 4340 | 4340 | 4340 | 4340 | 4340 | 4340 | 4340 | 4340 |

[1] The 'r' is short for Pearson product–moment correlation coefficient. [2] The 'N' is short for number of data. The number marked by * and ** means test value is less than 0.05 and 0.01, with significant correlation.

## Appendix D. Correlation between Sentiment Scores and Elements in Yokohama

| Column | | Sentiment | Sky | Plant | Water | Cloud | Flower | Wood | Pond | Food | Building | Bird | Place | Beach | People | Parking Lot | Sea | Walk | Temple | Pond | Park | Shrine |
|---|---|---|---|---|---|---|---|---|---|---|---|---|---|---|---|---|---|---|---|---|---|---|
| Sentiment | r [1] | 1 | −0.107 * | 0.132 * | 0.187 * | 0.037 | 0.034 ** | 0.039 | 0.008 | 0.031 | −0.139 ** | 0.029 | 0.030 | 0.035 | 0.064 | 0.086 * | 0.056 | 0.025 | 0.016 | 0.055 | 0.069 | 0.017 |
| | N [2] | 559 | 559 | 559 | 559 | 559 | 559 | 559 | 559 | 559 | 559 | 559 | 559 | 559 | 559 | 559 | 559 | 559 | 559 | 559 | 559 | 559 |
| Sky | r | −0.107 * | 1 | −0.316 ** | 0.078 | 0.177 ** | −0.220 ** | −0.016 | −0.102 * | −0.144 ** | 0.075 | −0.075 | 0.037 | 0.054 * | 0.010 | 0.008 | −0.029 | −0.025 | −0.033 | −0.061 | 0.030 | −0.110 ** |
| | N | 559 | 559 | 559 | 559 | 559 | 559 | 559 | 559 | 559 | 559 | 559 | 559 | 559 | 559 | 559 | 559 | 559 | 559 | 559 | 559 | 559 |
| Plant | r | 0.132 * | −0.316 ** | 1 | −0.302 ** | −0.310 ** | 0.197 ** | −0.128 ** | −0.082 | −0.087 * | −0.010 | −0.028 | 0.009 | −0.134 ** | −0.019 | −0.068 | −0.083 | 0.020 | 0.159 ** | 0.100 * | −0.027 | 0.178 ** |
| | N | 559 | 559 | 559 | 559 | 559 | 559 | 559 | 559 | 559 | 559 | 559 | 559 | 559 | 559 | 559 | 559 | 559 | 559 | 559 | 559 | 559 |
| Water | r | 0.187 * | 0.078 | −0.302 ** | 1 | 0.015 | −0.133 ** | 0.115 ** | −0.075 | −0.113 ** | −0.084 * | −0.011 | −0.055 | 0.067 | 0.012 | 0.017 | 0.055 | 0.089 * | −0.061 | −0.015 | −0.001 | −0.133 ** |
| | N | 559 | 559 | 559 | 559 | 559 | 559 | 559 | 559 | 559 | 559 | 559 | 559 | 559 | 559 | 559 | 559 | 559 | 559 | 559 | 559 | 559 |
| Cloud | r | 0.037 | 0.177 ** | −0.310 ** | 0.015 | 1 | −0.133 ** | 0.085 * | −0.041 | −0.086 * | −0.073 | −0.040 | −0.062 | 0.173 ** | 0.004 | 0.047 | 0.036 | −0.067 | −0.093 * | −0.043 | 0.062 | −0.116 ** |
| | N | 559 | 559 | 559 | 559 | 559 | 559 | 559 | 559 | 559 | 559 | 559 | 559 | 559 | 559 | 559 | 559 | 559 | 559 | 559 | 559 | 559 |
| Flower | r | 0.034 ** | −0.220 ** | 0.197 ** | −0.133 ** | −0.133 ** | 1 | −0.037 | −0.033 | −0.029 | −0.022 | 0.001 | −0.040 | −0.062 | 0.020 | −0.045 | −0.034 | 0.006 | −0.001 | −0.008 | −0.039 | 0.086 * |
| | N | 559 | 559 | 559 | 559 | 559 | 559 | 559 | 559 | 559 | 559 | 559 | 559 | 559 | 559 | 559 | 559 | 559 | 559 | 559 | 559 | 559 |
| Pond | r | 0.039 | −0.016 | −0.128 ** | 0.115 ** | 0.085 * | −0.037 | 1 | −0.049 | −0.032 | −0.012 | −0.011 | 0.006 | −0.008 | 0.029 | 0.042 | −0.008 | 0.059 | 0.002 | −0.043 | −0.047 | −0.055 |
| | N | 559 | 559 | 559 | 559 | 559 | 559 | 559 | 559 | 559 | 559 | 559 | 559 | 559 | 559 | 559 | 559 | 559 | 559 | 559 | 559 | 559 |
| Dog | r | 0.008 | −0.102 * | −0.082 | −0.075 | −0.041 | −0.033 | −0.049 | 1 | −0.018 | −0.021 | −0.016 | −0.038 | 0.019 | 0.084 * | 0.009 | −0.005 | 0.062 | −0.029 | −0.026 | −0.027 | −0.025 |
| | N | 559 | 559 | 559 | 559 | 559 | 559 | 559 | 559 | 559 | 559 | 559 | 559 | 559 | 559 | 559 | 559 | 559 | 559 | 559 | 559 | 559 |
| Food | r | 0.031 | −0.144 ** | −0.087 * | −0.113 ** | −0.086 * | −0.029 | −0.032 | −0.018 | 1 | −0.017 | −0.013 | −0.018 | −0.016 | 0.005 | −0.025 | 0.255 ** | −0.007 | −0.023 | −0.021 | −0.022 | −0.020 |
| | N | 559 | 559 | 559 | 559 | 559 | 559 | 559 | 559 | 559 | 559 | 559 | 559 | 559 | 559 | 559 | 559 | 559 | 559 | 559 | 559 | 559 |
| Building | r | −0.139 ** | 0.075 | −0.010 | −0.084 * | −0.073 | −0.022 | −0.012 | −0.021 | −0.017 | 1 | −0.014 | 0.046 | −0.036 | −0.035 | −0.028 | −0.024 | −0.017 | 0.010 | −0.006 | −0.025 | 0.096 * |
| | N | 559 | 559 | 559 | 559 | 559 | 559 | 559 | 559 | 559 | 559 | 559 | 559 | 559 | 559 | 559 | 559 | 559 | 559 | 559 | 559 | 559 |
| Bird | r | 0.029 | −0.075 | −0.028 | −0.011 | −0.040 | 0.001 | −0.011 | −0.016 | −0.013 | −0.014 | 1 | −0.001 | −0.013 | −0.004 | −0.010 | −0.014 | 0.000 | 0.076 | 0.074 | 0.008 | −0.018 |
| | N | 559 | 559 | 559 | 559 | 559 | 559 | 559 | 559 | 559 | 559 | 559 | 559 | 559 | 559 | 559 | 559 | 559 | 559 | 559 | 559 | 559 |
| place | r | 0.030 | 0.037 | 0.009 | −0.055 | −0.062 | −0.040 | 0.006 | −0.038 | −0.018 | 0.046 | −0.001 | 1 | −0.041 | 0.001 | −0.012 | −0.025 | −0.041 | 0.041 | −0.014 | −0.034 | −0.041 |
| | N | 559 | 559 | 559 | 559 | 559 | 559 | 559 | 559 | 559 | 559 | 559 | 559 | 559 | 559 | 559 | 559 | 559 | 559 | 559 | 559 | 559 |
| beach | r | 0.035 | 0.054 * | −0.134 ** | 0.067 | 0.173 ** | −0.062 | −0.008 | 0.019 | −0.016 | −0.036 | −0.013 | −0.041 | 1 | 0.003 | −0.015 | 0.019 | −0.035 | −0.050 | −0.046 | 0.047 | −0.044 |
| | N | 559 | 559 | 559 | 559 | 559 | 559 | 559 | 559 | 559 | 559 | 559 | 559 | 559 | 559 | 559 | 559 | 559 | 559 | 559 | 559 | 559 |
| people | r | 0.064 | 0.010 | −0.019 | 0.012 | 0.004 | 0.020 | 0.029 | 0.084 * | 0.005 | −0.035 | −0.004 | 0.001 | 0.003 | 1 | 0.070 | 0.019 | −0.008 | −0.002 | −0.029 | −0.005 | −0.040 |
| | N | 559 | 559 | 559 | 559 | 559 | 559 | 559 | 559 | 559 | 559 | 559 | 559 | 559 | 559 | 559 | 559 | 559 | 559 | 559 | 559 | 559 |
| parking lot | r | 0.086 * | 0.008 | −0.068 | 0.017 | 0.047 | −0.045 | 0.042 | 0.009 | −0.025 | −0.028 | −0.010 | −0.012 | −0.015 | 0.070 | 1 | 0.021 | −0.017 | −0.036 | −0.031 | −0.004 | −0.026 |
| | N | 559 | 559 | 559 | 559 | 559 | 559 | 559 | 559 | 559 | 559 | 559 | 559 | 559 | 559 | 559 | 559 | 559 | 559 | 559 | 559 | 559 |
| sea | r | 0.056 | −0.029 | −0.083 | 0.055 | 0.036 | −0.034 | −0.008 | −0.005 | 0.255 ** | −0.024 | −0.014 | −0.025 | 0.019 | 0.019 | 0.021 | 1 | 0.019 | −0.035 | −0.033 | −0.015 | −0.016 |
| | N | 559 | 559 | 559 | 559 | 559 | 559 | 559 | 559 | 559 | 559 | 559 | 559 | 559 | 559 | 559 | 559 | 559 | 559 | 559 | 559 | 559 |
| walk | r | 0.025 | −0.025 | 0.020 | 0.089 * | −0.067 | 0.006 | 0.059 | 0.062 | −0.007 | −0.017 | 0.000 | −0.041 | −0.035 | −0.008 | −0.017 | 0.019 | 1 | −0.021 | 0.019 | −0.003 | −0.011 |
| | N | 559 | 559 | 559 | 559 | 559 | 559 | 559 | 559 | 559 | 559 | 559 | 559 | 559 | 559 | 559 | 559 | 559 | 559 | 559 | 559 | 559 |
| temple | r | 0.016 | −0.033 | 0.159 ** | −0.061 | −0.093 * | −0.001 | 0.002 | −0.029 | −0.023 | 0.010 | 0.076 | 0.041 | −0.050 | −0.002 | −0.036 | −0.035 | −0.021 | 1 | 0.226 ** | 0.019 | −0.032 |
| | N | 559 | 559 | 559 | 559 | 559 | 559 | 559 | 559 | 559 | 559 | 559 | 559 | 559 | 559 | 559 | 559 | 559 | 559 | 559 | 559 | 559 |
| pond | r | 0.055 | −0.061 | 0.100 * | −0.015 | −0.043 | −0.008 | −0.043 | −0.026 | −0.021 | −0.006 | 0.074 | −0.014 | −0.046 | −0.029 | −0.031 | −0.033 | 0.019 | 0.226 ** | 1 | −0.028 | −0.029 |
| | N | 559 | 559 | 559 | 559 | 559 | 559 | 559 | 559 | 559 | 559 | 559 | 559 | 559 | 559 | 559 | 559 | 559 | 559 | 559 | 559 | 559 |
| park | r | 0.069 | 0.030 | −0.027 | −0.001 | 0.062 | −0.039 | −0.047 | −0.027 | −0.022 | −0.025 | 0.008 | −0.034 | 0.047 | −0.005 | −0.004 | −0.015 | −0.003 | 0.019 | −0.028 | 1 | −0.031 |
| | N | 559 | 559 | 559 | 559 | 559 | 559 | 559 | 559 | 559 | 559 | 559 | 559 | 559 | 559 | 559 | 559 | 559 | 559 | 559 | 559 | 559 |
| shrine | r | 0.017 | −0.110 ** | 0.178 ** | −0.133 ** | −0.116 ** | 0.086 * | −0.055 | −0.025 | −0.020 | 0.096 * | −0.018 | −0.041 | −0.044 | −0.040 | −0.026 | −0.016 | −0.011 | −0.032 | −0.029 | −0.031 | 1 |
| | N | 559 | 559 | 559 | 559 | 559 | 559 | 559 | 559 | 559 | 559 | 559 | 559 | 559 | 559 | 559 | 559 | 559 | 559 | 559 | 559 | 559 |

[1] The 'r' is short for Pearson product–moment correlation coefficient. [2] The 'N' is short for number of data. The number marked by * and ** means test value is less than 0.05 and 0.01, with significant correlation.

## Appendix E. Correlation between Sentiment Scores and Elements in Otsu

| Column | | Sentiment | Plant | Sky | Cloud | Flower | Water | Wood | Food | Pond | Building | Tableware | Temple | Place | Lake Biwa | Autumn Leaves | Parking Lot | Hall | Murasaki Shikibu | Precincts | People | Cherry Blossoms |
|---|---|---|---|---|---|---|---|---|---|---|---|---|---|---|---|---|---|---|---|---|---|---|
| Sentiment | r [1] | 1 | 0.063 ** | −0.001 | −0.013 | 0.052 ** | −0.025 | 0.061 ** | 0.065 ** | 0.001 | 0.005 | 0.038 | 0.015 | 0.004 | −0.001 | −0.010 | 0.052 ** | 0.036 | −0.020 | 0.059 ** | 0.053 ** | 0.022 |
| | N [2] | 2462 | 2462 | 2462 | 2462 | 2462 | 2462 | 2462 | 2462 | 2462 | 2462 | 2462 | 2462 | 2462 | 2462 | 2462 | 2462 | 2462 | 2462 | 2462 | 2462 | 2462 |
| Plant | r | 0.063 ** | 1 | −0.298 ** | −0.223 ** | 0.005 | −0.271 ** | −0.076 ** | −0.065 ** | −0.047 * | −0.013 | −0.054 ** | 0.027 | −0.028 | −0.095 ** | 0.091 ** | 0.031 | 0.008 | 0.074 ** | 0.053 ** | −0.004 | 0.034 * |
| | N | 2462 | 2462 | 2462 | 2462 | 2462 | 2462 | 2462 | 2462 | 2462 | 2462 | 2462 | 2462 | 2462 | 2462 | 2462 | 2462 | 2462 | 2462 | 2462 | 2462 | 2462 |
| Sky | r | −0.001 | −0.298 ** | 1 | 0.145 ** | −0.143 ** | 0.095 ** | −0.115 ** | −0.084 ** | −0.009 | 0.019 | −0.079 ** | −0.080 ** | 0.007 | 0.086 ** | −0.074 ** | −0.002 | 0.064 ** | −0.058 ** | 0.042 * | −0.048 * | −0.035 |
| | N | 2462 | 2462 | 2462 | 2462 | 2462 | 2462 | 2462 | 2462 | 2462 | 2462 | 2462 | 2462 | 2462 | 2462 | 2462 | 2462 | 2462 | 2462 | 2462 | 2462 | 2462 |
| Cloud | r | −0.013 | −0.223 ** | 0.145 ** | 1 | −0.108 ** | 0.044 * | −0.076 ** | −0.060 ** | 0.030 | −0.018 | −0.039 | −0.046 * | 0.035 | 0.133 ** | −0.103 ** | −0.028 | −0.032 | −0.043 * | −0.007 | 0.007 | −0.048 * |
| | N | 2462 | 2462 | 2462 | 2462 | 2462 | 2462 | 2462 | 2462 | 2462 | 2462 | 2462 | 2462 | 2462 | 2462 | 2462 | 2462 | 2462 | 2462 | 2462 | 2462 | 2462 |
| Flower | r | 0.052 ** | 0.005 | −0.143 ** | −0.108 ** | 1 | −0.171 ** | −0.031 | −0.040 * | −0.064 ** | 0.036 | −0.033 | −0.054 ** | −0.009 | −0.042 * | 0.005 | 0.051 * | −0.058 ** | −0.042 * | −0.042 * | 0.012 | 0.177 ** |
| | N | 2462 | 2462 | 2462 | 2462 | 2462 | 2462 | 2462 | 2462 | 2462 | 2462 | 2462 | 2462 | 2462 | 2462 | 2462 | 2462 | 2462 | 2462 | 2462 | 2462 | 2462 |
| Water | r | −0.025 | −0.271 ** | 0.095 ** | 0.044 * | −0.171 ** | 1 | −0.049 * | −0.060 ** | 0.170 ** | −0.010 | −0.055 ** | −0.046 * | 0.036 | 0.126 ** | −0.075 ** | 0.013 | −0.014 | −0.072 ** | −0.039 | −0.021 | −0.049 * |
| | N | 2462 | 2462 | 2462 | 2462 | 2462 | 2462 | 2462 | 2462 | 2462 | 2462 | 2462 | 2462 | 2462 | 2462 | 2462 | 2462 | 2462 | 2462 | 2462 | 2462 | 2462 |
| Wood | r | 0.061 ** | −0.076 ** | −0.115 ** | −0.076 ** | −0.031 | −0.049 * | 1 | −0.002 | −0.002 | −0.010 | −0.020 | −0.019 | −0.020 | −0.027 | 0.035 | −0.015 | −0.007 | −0.022 | −0.005 | 0.103 ** | −0.019 |
| | N | 2462 | 2462 | 2462 | 2462 | 2462 | 2462 | 2462 | 2462 | 2462 | 2462 | 2462 | 2462 | 2462 | 2462 | 2462 | 2462 | 2462 | 2462 | 2462 | 2462 | 2462 |
| Food | r | 0.065 ** | −0.065 ** | −0.084 ** | −0.060 ** | −0.040 * | −0.060 ** | −0.002 | 1 | −0.025 | −0.023 | 0.696 ** | 0.020 | −0.028 | −0.031 | 0.001 | −0.009 | 0.020 | −0.030 | 0.020 | −0.023 | 0.017 ** |
| | N | 2462 | 2462 | 2462 | 2462 | 2462 | 2462 | 2462 | 2462 | 2462 | 2462 | 2462 | 2462 | 2462 | 2462 | 2462 | 2462 | 2462 | 2462 | 2462 | 2462 | 2462 |
| Pond | r | 0.001 | −0.047 * | −0.009 | 0.030 | −0.064 ** | 0.170 ** | −0.002 | −0.025 | 1 | 0.013 | −0.022 | −0.002 | 0.012 | 0.082 ** | −0.028 | −0.022 | −0.020 | 0.008 | 0.016 | −0.017 | −0.007 |
| | N | 2462 | 2462 | 2462 | 2462 | 2462 | 2462 | 2462 | 2462 | 2462 | 2462 | 2462 | 2462 | 2462 | 2462 | 2462 | 2462 | 2462 | 2462 | 2462 | 2462 | 2462 |
| Building | r | 0.005 | −0.013 | 0.019 | −0.018 | 0.036 | −0.010 | −0.010 | −0.023 | 0.013 | 1 | −0.020 | −0.031 | −0.009 | −0.006 | −0.027 | −0.004 | −0.022 | −0.022 | 0.000 | −0.018 | 0.034 |
| | N | 2462 | 2462 | 2462 | 2462 | 2462 | 2462 | 2462 | 2462 | 2462 | 2462 | 2462 | 2462 | 2462 | 2462 | 2462 | 2462 | 2462 | 2462 | 2462 | 2462 | 2462 |
| Tableware | r | 0.038 | −0.054 ** | −0.079 ** | −0.039 | −0.033 | −0.055 ** | −0.020 | 0.696 ** | −0.022 | −0.020 | 1 | 0.009 | −0.024 | −0.025 | 0.010 | −0.008 | 0.022 | −0.026 | 0.036 | −0.020 | 0.037 |
| | N | 2462 | 2462 | 2462 | 2462 | 2462 | 2462 | 2462 | 2462 | 2462 | 2462 | 2462 | 2462 | 2462 | 2462 | 2462 | 2462 | 2462 | 2462 | 2462 | 2462 | 2462 |
| temple | r | 0.015 | 0.027 | −0.080 ** | −0.046 * | −0.054 ** | −0.046 * | −0.019 | 0.020 | −0.002 | −0.031 | 0.009 | 1 | −0.045 * | 0.000 | −0.045 * | −0.031 | −0.007 | 0.076 ** | −0.006 | −0.018 | −0.011 |
| | N | 2462 | 2462 | 2462 | 2462 | 2462 | 2462 | 2462 | 2462 | 2462 | 2462 | 2462 | 2462 | 2462 | 2462 | 2462 | 2462 | 2462 | 2462 | 2462 | 2462 | 2462 |
| place | r | 0.004 | −0.028 | 0.007 | 0.035 | −0.009 | 0.036 | −0.020 | −0.028 | 0.012 | −0.009 | −0.024 | −0.045 * | 1 | −0.018 | −0.036 | −0.036 | −0.028 | 0.028 | −0.015 | 0.009 | 0.002 |
| | N | 2462 | 2462 | 2462 | 2462 | 2462 | 2462 | 2462 | 2462 | 2462 | 2462 | 2462 | 2462 | 2462 | 2462 | 2462 | 2462 | 2462 | 2462 | 2462 | 2462 | 2462 |
| lake Biwa | r | −0.001 | −0.095 ** | 0.086 ** | 0.133 ** | −0.042 * | 0.126 ** | −0.027 | −0.031 | 0.082 ** | −0.006 | −0.025 | 0.000 | −0.018 | 1 | −0.041 * | −0.018 | 0.010 | −0.040 * | 0.045 * | −0.022 | −0.017 |
| | N | 2462 | 2462 | 2462 | 2462 | 2462 | 2462 | 2462 | 2462 | 2462 | 2462 | 2462 | 2462 | 2462 | 2462 | 2462 | 2462 | 2462 | 2462 | 2462 | 2462 | 2462 |
| autumn leaves | r | −0.010 | 0.091 ** | −0.074 ** | −0.103 ** | 0.005 | −0.075 ** | 0.035 | 0.001 | −0.028 | −0.027 | 0.010 | −0.045 * | −0.036 | −0.041 * | 1 | −0.029 | −0.019 | −0.017 | −0.029 | −0.017 | −0.015 |
| | N | 2462 | 2462 | 2462 | 2462 | 2462 | 2462 | 2462 | 2462 | 2462 | 2462 | 2462 | 2462 | 2462 | 2462 | 2462 | 2462 | 2462 | 2462 | 2462 | 2462 | 2462 |
| parking lot | r | 0.052 ** | 0.031 | −0.002 | −0.028 | 0.051 * | 0.013 | −0.015 | −0.009 | −0.022 | −0.004 | −0.008 | −0.031 | −0.036 | −0.018 | −0.029 | 1 | −0.016 | −0.025 | 0.042 * | −0.018 | −0.004 |
| | N | 2462 | 2462 | 2462 | 2462 | 2462 | 2462 | 2462 | 2462 | 2462 | 2462 | 2462 | 2462 | 2462 | 2462 | 2462 | 2462 | 2462 | 2462 | 2462 | 2462 | 2462 |
| hall | r | 0.036 | 0.008 | 0.064 ** | −0.032 | −0.058 ** | −0.014 | −0.007 | 0.020 | −0.020 | −0.022 | 0.022 | −0.007 | −0.028 | 0.010 | −0.019 | −0.016 | 1 | 0.003 | 0.099 ** | −0.016 | −0.022 |
| | N | 2462 | 2462 | 2462 | 2462 | 2462 | 2462 | 2462 | 2462 | 2462 | 2462 | 2462 | 2462 | 2462 | 2462 | 2462 | 2462 | 2462 | 2462 | 2462 | 2462 | 2462 |
| murasaki Shikibu | r | −0.020 | 0.074 ** | −0.058 ** | −0.043 * | −0.042 * | −0.072 ** | −0.022 | −0.030 | 0.008 | −0.022 | −0.026 | 0.076 ** | 0.028 | −0.040 * | −0.017 | −0.025 | 0.003 | 1 | 0.006 | −0.026 | −0.024 |
| | N | 2462 | 2462 | 2462 | 2462 | 2462 | 2462 | 2462 | 2462 | 2462 | 2462 | 2462 | 2462 | 2462 | 2462 | 2462 | 2462 | 2462 | 2462 | 2462 | 2462 | 2462 |
| precincts | r | 0.059 ** | 0.053 ** | 0.042 * | −0.007 | −0.042 * | −0.039 | −0.005 | 0.020 | 0.016 | 0.000 | 0.036 | −0.006 | −0.015 | 0.045 * | −0.029 | 0.042 * | 0.099 ** | 0.006 | 1 | −0.016 | 0.029 |
| | N | 2462 | 2462 | 2462 | 2462 | 2462 | 2462 | 2462 | 2462 | 2462 | 2462 | 2462 | 2462 | 2462 | 2462 | 2462 | 2462 | 2462 | 2462 | 2462 | 2462 | 2462 |
| people | r | 0.053 ** | −0.004 | −0.048 * | 0.007 | 0.012 | −0.021 | 0.103 ** | −0.023 | −0.017 | −0.018 | −0.020 | −0.018 | 0.009 | −0.022 | −0.017 | −0.018 | −0.016 | −0.026 | −0.016 | 1 | −0.019 |
| | N | 2462 | 2462 | 2462 | 2462 | 2462 | 2462 | 2462 | 2462 | 2462 | 2462 | 2462 | 2462 | 2462 | 2462 | 2462 | 2462 | 2462 | 2462 | 2462 | 2462 | 2462 |
| cherry blossoms | r | 0.022 | 0.034 * | −0.035 | −0.048 * | 0.177 ** | −0.049 * | −0.019 | 0.017 ** | −0.007 | 0.034 | 0.037 | −0.011 | 0.002 | −0.017 | −0.015 | −0.004 | −0.022 | −0.024 | 0.029 | −0.019 | 1 |
| | N | 2462 | 2462 | 2462 | 2462 | 2462 | 2462 | 2462 | 2462 | 2462 | 2462 | 2462 | 2462 | 2462 | 2462 | 2462 | 2462 | 2462 | 2462 | 2462 | 2462 | 2462 |

[1] The 'r' is short for Pearson product–moment correlation coefficient. [2] The 'N' is short for number of data. The number marked by * and ** means test value is less than 0.05 and 0.01, with significant correlation.

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
