# Peer review of "Cross-Cultural Comparison of Urban Green Space through Crowdsourced Big Data: A Natural Language Processing and Image Recognition Approach"

_land, doi:10.3390/land12040767_

Round 1

Reviewer 1 Report

Dear Authors,

The concept of the work is interesting and the goal is right. However, the work needs several corrections.

Regarding the Introduction

a.  All research serves some goals. What is the aim of your work? It should be stated in the Introduction. You mentioned the practical value of your work at the end of the Conclusions, but without describing the goals from the beginning, it is not very convincing. 

b. The content of the introduction is not enough, and its literature review should be enriched. The current content shows many related work achievements that have not been cited.  How can the current research findings contribute to crucial for UGS design and management? What's the point of this?

Regarding the Materials and Methods

a.  A flow chart to describe the procedure of the experiment is needed. 

b.  Lines 118-158. Why add so much background to the method? Isn't this supposed to be stated in the introduction?

Regarding the Results:

a. The results are quite chaotic. The readability of the results should be improved. A large number of long tables greatly reduce the continuity of reading.

b. Tables and table names should be corrected to be more correct and legible.

Regarding the Discussion and Conclusion

a. What is the innovation of the research? Where is your contribution?

Regarding the Limitations

In the limitations of your research, you mentioned that social media data may not fully understand urban green space. So can you consider the content mentioned above to be useless and unreliable?

Both language and organization quality should be improved.  So it is obvious that proofreading should be conducted by a native English speaker

I encourage you to revise your work so that it has a chance to be published as I find your proposal interesting.

Best regards.

Author Response

Dear reviewer,

Thank you for your valuable comments. I have carefully considered each of your comments and revised the manuscript accordingly. Please find the details of the revisions in the PDF file that I have uploaded for your reference.

Yours,

Shuhao Liu

Reviewer 2 Report

This paper evaluates the public's perceived attitudes toward UGS in multiple locations, using NLP and Computer Vision methods. Overall, the paper is well-written and presents some good arguments.

1- The abstract is well-written and provides a good summary of the paper. Perhaps a bit more can be said about the methodology used.

2- In the introduction, make clearer what knowledge gaps you identified and how your research addresses them. Also, make the research objectives/questions clearer. Answer the “so what?” question. Why investigating such matter is important? End the introduction with an outline of the paper; what comes next?

3- The novelty/originality should be clearly justified that the manuscript contains sufficient contributions to the new body of knowledge from the international perspective.  What new things (new theories, new methods, or new policies) can the paper contribute to the existing international literature? This point must be reasonably justified by a Literature Review, clearly introduced in Introduction Section, and completely discussed in Discussion Section.

4- you need a new section on literature review. You need to acknowledge the existing literature on the issue and clearly identify the knowledge gap.

5- recently an special issue of a similar topic has been published which I suggest the authors to have a look: https://www.frontiersin.org/articles/10.3389/fenvs.2022.1050597/full there are multiple paper in this special issue which might be of interest to this paper.

6- your discussion is premature and needs further elaboration. More work is needed to substantiate the discussions in your manuscript.

7- The conclusion could do more to tease out the wider resonance of the paper for the journal's international readership.

8- What are the limitations of your study? This can be added at the end of the methodology section.

9- you need to refer back to the literature and previous studies in your result, discussion and conclusion sections.

10- how generalisable your findings are to other places? Provide some discussions around the generalisability of your findings in the discussion section.

Author Response

(The authors gave the same response as above.)

Reviewer 3 Report

I highly appreciate the importance of this comparative study on urban green space perceptions in two Asiatic countries. Both countries have a great green space culture, and good results obtained by similar or different management experiences.  Using a rich and an appropriate methodology, the authors offer to the readers a real model for further development of their research.

Please find below my comments and suggestions:

a)      The Title of the paper reflects its content and is attractive to prospective readers. Mentioning just in title the main methodological tools, the scholars will carefully look and analyze the paper. I have only a small suggestion for the title, which could be: “Cross-Cultural Comparison of Urban Green Space through Crowdsourced…..”. I added Urban and I deleted Utilization!

b)       The Abstract is relevant for the readers covering the departure point of analysis, the findings, and the importance of such studies, which should be developed.

c)       The Introduction seems to be well conceived, synthetically presenting the main research questions, and highlights the contribution of the study to international knowledge flow. I feel the necessity to introduce the second phrase in this section to specify that in your paper, for facilitate the analysis, you have used the urban green space (UGS) including the urban blue space (UBS). Usually, the scholars make distinction between both categories.

Please, reformulate the phrase between rows 34 and 36 to eliminate the annoying repetition of „UGS”.

d)    The Materials and Methods section has an appropriate structure, describing the study areas, mentioning the key data providers, the relevance of data collected and the appropriate used methods. However, I suggest reflecting on a clear grouping of the contents in two subsections: Study areas and data collection (first two current subchapters) and Methods (the last three subsections). I appreciate the basic geographical information for each study area, which facilitate the understanding of the specificity of the places, and synthetical description of each applied methods.

e)      With its three sub-sections, the Results represents a relevant findings’ synthesis, creating an image about the public perceptions of UGS. A detailed statistical analysis is presented for four selected case studies (Guangzhou, Changsha, Yokohama and Otsu), showing the correlation level between sentiment scores and different elements depicted in the previous pages. Regarding this analysis, I suggest moving all these tables (starting with Table 3 until 6) as Appendixes, because is more relevant the conclusions for each, and less a lot of figures. In addition, this mass of data shifts the reader’s focus from ideas to numbers!

f)        Discussion section is an excellent comment and starting debatable points on obtained results, with interesting remarks on the comparative analysis of the affective scores of the Chinese and  Japanese UGSs. The specific analysis of environmental characteristics of UGS in the selected cities, for each country, reveals the specific of places, which can help the local authorities to improve their urban planning design.

I suggest adding of the 5.1. subchapter from the next section, at the end of Discussion, considering that this is the best place (as 4.3 subchapter).

g)       Conclusion, as the last section of the paper, is a short by relevant synthesis on the main ideas.

If the authors decide to do not move the subchapter Limitations and Future Research, I suggest inserting it in the text, but not as a separate subchapter. Is totally unusually to have a section with only one subsection. Clearly underlying the limitations of study, it is an excellent encouraging for researchers to focus their future research on the weak points highlighted by the authors themselves.

To conclude, in my opinion, this paper is required by existing literature in the field, developing new methodological ways applied in the comparative analysis of the public perception of the UGS design in different cultural and environmental conditions. This study could be seen as a modality to improve the resilient capacity of cities by an optimal valorization of urban green space potential.  

Author Response

(The authors gave the same response as above.)

Round 2

Reviewer 1 Report

I recommend accepting it in its current form.

Reviewer 2 Report

Thanks for addressing the comments.